# *MyoD1* localization at the nuclear periphery is mediated by association of WFS1 with active enhancers

Konstantina Georgiou[1,2,3], Fatih Sarigol [1,2], Tobias Nimpf[1,2], Christian Knapp [1,2,4], Daria Filipczak[1,2,3], Roland Foisner [1,2,5] & Nana Naetar [1,2,5]

Spatial organization of the mammalian genome influences gene expression and cell identity. While association of genes with the nuclear periphery is commonly linked to transcriptional repression, also active, expressed genes can localize at the nuclear periphery. The transcriptionally active *MyoD1* gene, a master regulator of myogenesis, exhibits peripheral localization in proliferating myoblasts, yet the underlying mechanisms remain elusive. Here, we generate a reporter cell line to demonstrate that peripheral association of the *MyoD1* locus is independent of mechanisms involved in heterochromatin anchoring. Instead, we identify the nuclear envelope transmembrane protein WFS1 that tethers *MyoD1* to the nuclear periphery. WFS1 primarily associates with active distal enhancer elements upstream of *MyoD1*, and with a subset of enhancers genome-wide, which are enriched in active histone marks and linked to expressed myogenic genes. Overall, our data identify a mechanism involved in tethering regulatory elements of active genes to the nuclear periphery.

Spatial organization of the mammalian genome is a non-random process that contributes to the coordinated regulation of gene expression[1–6], however the molecular mechanisms involved are only partially understood.

At a higher-order organizational level, genomes assemble into A and B compartments corresponding to euchromatin and heterochromatin, respectively. Microscopic studies[7,8] and genome-wide analyses, such as chromosome conformation capture and DamID techniques[9,10] revealed the spatial segregation of these compartments, with the B compartment mainly located at the nuclear periphery and the A compartment enriched in the nuclear interior[9,11,12]. This three-dimensional organization of the genome is, in part, mediated by association of chromatin with structural elements of the nucleus such as nuclear bodies in the nuclear interior[13–15] and the

nuclear lamina, a structural scaffold located beneath the inner nuclear membrane (INM)[16–18]. Two types of lamins, A- and B-type, with distinct expression patterns and biochemical properties[19] are the main components of the lamina. They interact with chromatin either directly or indirectly through nuclear envelope transmembrane proteins (NETs)[20,21]. Genomic regions associated with the nuclear lamina are termed lamina-associated domains (LADs) and cover more than a third of the human genome[22]. LADs are large, gene-poor heterochromatic regions, enriched in the repressive histone modifications H3K9me2/3 and H3K27me3[22,23]. Anchoring of heterochromatic LADs at the nuclear periphery is mediated by two redundant mechanisms, involving lamin B receptor (LBR)[24] and complexes of A-type lamins with specific NETs[25]. LBR, a lamin B-binding INM protein tethers heterochromatin to the membrane directly via its Tudor

[1]Max Perutz Labs, Vienna Biocenter Campus (VBC), Vienna, Austria. [2]Max Perutz Labs, Medical University of Vienna, Vienna, Austria. [3]Vienna BioCenter PhD Program, a Doctoral School of the University of Vienna and Medical University of Vienna, Vienna, Austria. [4]Present address: ICFO-Institut de Ciencies Fotoniques, The Barcelona Institute of Science and Technology, Castelldefels, Barcelona, Spain. [5]These authors contributed equally: Roland Foisner, Nana Naetar. ✉e-mail: roland.foisner@meduniwien.ac.at; nana.naetar@meduniwien.ac.at

domain[26], or indirectly by binding to heterochromatin protein 1 (HP1)[27]. A-type lamins anchor heterochromatin at the nuclear periphery mostly indirectly via a group of lamin-binding INM proteins containing a LAP-Emerin-MAN1 (LEM) domain[28]. The LEM domain binds to DNA-associated Barrier-to-Autointegration Factor[29] mediating the interaction of these proteins with chromatin. Certain LEM proteins, such as emerin and LAP2β also bind histone deacetylase HDAC3 further contributing to peripheral heterochromatin localization[5,30,31]. In mammals, additional proteins seem to facilitate peripheral LAD anchorage, for instance, PRR14 through lamin A/C and HP1 binding[32,33], as well as the lamin B-interacting proteins Prdm16[34] and ZKSCAN3[35]. In *C. elegans*, chromo-domain protein Cec-4 tethers heterochromatin to the nuclear periphery through binding to histone 3 methylated at lysine 9 (H3K9me2/3)[2,36,37].

Peripheral heterochromatin is known to form a repressive environment, as most genes within LADs are repressed[22]. In addition, several studies found that association of genes with the nuclear periphery leads to transcriptional repression, while detachment from the nuclear lamina correlates with their activation[38–42]. Despite the overall repressive environment of heterochromatin at the nuclear periphery, active genes can also be located at the periphery, but the mechanism of how they are tethered and maintained in an active state is poorly understood. A systematic approach revealed a subset of promoters and enhancers located within LADs that can escape the repressive environment and promote transcription[43]. In addition, artificial tethering of genomic loci to the nuclear lamina yielded inconsistent results[44–46], showing that genes can escape the repressive environment under specific conditions that are not yet fully understood. Moreover, peripheral chromatin domains, termed H3K9me2-Only Domains (KODs), which are epigenetically distinct from LADs and contain accessible chromatin and distal enhancers, were found to associate with the nuclear membrane in a lamin B-independent manner[47]. Finally, certain peripherally anchored genes were shown to be released but spatially confined in the proximity of the lamina upon their activation[48]. Collectively, these findings provide evidence for active chromatin regulation both within and outside LAD regions at the nuclear periphery, but little is known about the tethers and mechanisms of activation.

A prominent example of a peripherally positioned, yet active gene is myogenic determination gene number 1 (*MyoD1*)[49], encoding the MyoD1 protein, a master regulator of skeletal myogenesis. In proliferating myoblasts, the *MyoD1* locus is found within an inter-LAD (iLAD) region and localizes at the nuclear periphery, where it is expressed at basal levels. During differentiation, it translocates towards the nuclear interior, coinciding with its up-regulation[50], and induces the expression of downstream myogenic genes required for terminal muscle cell differentiation[51,52]. This spatiotemporal regulation of *MyoD1* is crucial for myoblast function and differentiation. The INM protein emerin and histone deacetylase HDAC3 were proposed to mediate the transient peripheral attachment of *MyoD1* in myoblasts[53]. However, considering the dynamic and diverse protein composition of the INM[21] and the complexity of cell type-specific spatial and temporal genome organization within the nucleus, other NE components are likely to be involved.

In the present study, we investigate how *MyoD1* is tethered to the nuclear periphery in proliferating myoblasts. We generated a myoblast reporter cell line allowing easy detection of *MyoD1* 3D nuclear positioning in living cells, and find that neither peripheral tethers known to mediate anchorage of heterochromatin at the INM nor the heterochromatic environment itself are involved in attaching the active *MyoD1* locus to the nuclear periphery. Instead, WFS1, a NET that was previously shown to mediate the positioning of entire chromosomes to the nuclear periphery in muscle cells[40], is required for the peripheral anchorage of *MyoD1* in proliferating myoblasts. WFS1 binds to active distal enhancer elements upstream of the *MyoD1* gene and other genes, likely mediating their attachment to the nuclear periphery.

Thus, our study reveals a mechanism of active tethering of the expressed *MyoD1* gene to the periphery, mediated by WFS1 through association with the *MyoD1* enhancer region. As WFS1 also binds to numerous other epigenetically active enhancers predicted to regulate myogenic genes, it is tempting to speculate that this is a more general mechanism for positioning active regulatory elements at the nuclear periphery in myoblasts.

## Results

### Reporter cell system monitors the nuclear position of the *MyoD1* locus

To monitor the spatial position and regulation of *MyoD1* in single cells, we established a C2C12 myoblast reporter cell system by inserting a short array of Lac Operator (LacO) repeats 1 kb downstream of the last exon of *MyoD1* using CRISPR/Cas9 technology (Supplementary Fig. 1a-d). LacO repeats were visualized by stably expressing the Lac repressor (LacR)-GFP fusion protein (Fig. 1a). Myoblasts with properly integrated LacO repeats displayed 1-3 fluorescent dots in fluorescence and live microscopic analyses corresponding to the *MyoD1* locus (Fig. 1b, c, Supplementary Movie 1). As a control, we also generated a C2C12 myoblast line in which the LacO array was integrated downstream of the *Pax7* locus, previously shown to localize in the nuclear interior in proliferating myoblasts[53] (Supplementary Fig. 1b-e, Supplementary Movie 2). Single cell clones capable of undergoing myoblast differentiation were selected for subsequent experiments (referred to as *MyoD1* and *Pax7* LacO) (Supplementary Fig. 1f-h).

To systematically assess the spatial positioning of *MyoD1* and *Pax7* in nuclei, we devised a pipeline for semi-automated image acquisition and analysis, by combining multi-location high-throughput 3D image capture with a custom-built analysis pipeline to define the shortest radial distance of the tagged gene loci to the nuclear periphery (Supplementary Fig. 2a, see methods for details). In brief, image segmentation and demarcation of the nuclear border were done in the Hoechst channel, and gene loci were identified in the GFP channel using maximal projections in combination with a local maxima algorithm. Intranuclear spatial positions of gene loci were determined by assessing the 3D coordinates of the shortest vector from the gene spot to the nuclear border followed by normalization of the measured radial distance to the maximum distance possible based on the dimension of cells in z (for details see methods, Fig. 1d). Accuracy of gene spot detection by this semi-automated pipeline was confirmed through cross-comparison of analyzing visually detected signals versus signals identified using the image analysis pipeline (Supplementary Fig. 2b), revealing a false positive rate of 5% and a false negative rate of 19%. The reproducibility of automated loci detection was demonstrated through low variability in cumulative distribution functions derived from four independent replicate experiments (Supplementary Fig. 2c).

We determined and plotted the shortest radial distance of *MyoD1* and *Pax7* loci to the nuclear border in proliferating and differentiating myoblasts in density distribution and violin plots (Fig. 1e, f). The *MyoD1* locus relocated from the nuclear periphery towards the nuclear interior in differentiating myotubes (Fig. 1e, f), as previously demonstrated[50,53,54]. Moreover, and in accordance with earlier publications[53], the *Pax7* locus preferentially occupied a more central position within the nuclei of proliferating cells compared to *MyoD1* (Fig. 1e, f). We confirmed that LacO-tagging of the *MyoD1* gene did not interfere with its intranuclear positioning by visualizing the gene in control (untagged) and tagged myoblasts using a fluorescently labeled FISH probe for the *MyoD1* locus (Supplementary Fig. 2d). To exclude that potential changes in nuclear size influence gene position analyses, we showed that the mean nuclear height (z-dimension) of the tested cells did not change significantly during differentiation (Supplementary Fig. 2e), and that plotting of the absolute (non-normalized) radial distance of the loci to the nuclear periphery yielded results consistent with the normalized data (Supplementary Fig. 2f).

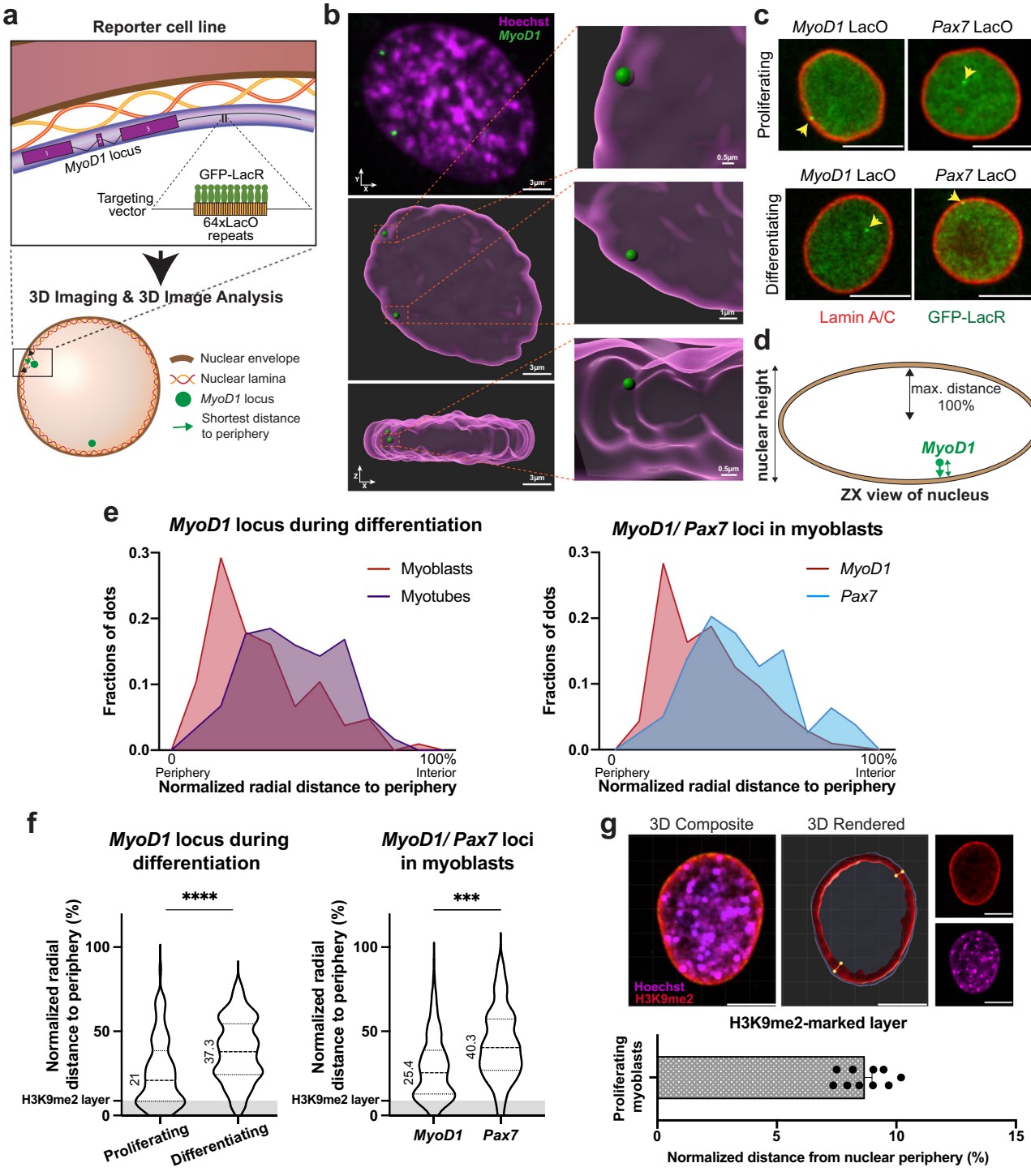

To confirm that the peripherally localized *MyoD1* loci, including those adjacent to but not directly at the nuclear periphery, were embedded in the peripheral heterochromatic layer, we additionally stained for the heterochromatic histone H3 lysine 9 dimethylation (H3K9me2) mark, which was previously demonstrated to specifically label heterochromatic regions at the nuclear periphery resembling heterochromatic LADs[38], and measured the thickness of the H3K9me2-marked layer (Fig. 1g and Supplementary Fig. 2g). Notably, the majority of *MyoD1* loci close to the nuclear periphery was indeed positioned within the H3K9me2-positive heterochromatic layer, as indicated by the gray area in the violin plots (Fig. 1f and Supplementary Fig. 2f).

In summary, we developed a robust and reliable cell reporter system that, coupled with image acquisition and analysis workflow,

facilitates efficient tracking of the intranuclear position of the *MyoD1* and *Pax7* genes.

## Heterochromatin reorganization does not affect *MyoD1* localization

We used the developed *MyoD1* reporter cell system to identify potential mechanisms involved in the peripheral positioning of *MyoD1* in proliferating myoblasts. First, we investigated whether the heterochromatic environment at the nuclear periphery determines *MyoD1* nuclear localization. To address this question, we deleted candidate genes known to be involved in anchoring heterochromatin to the periphery, including lamin A/C and the nuclear envelope transmembrane proteins emerin, LAP2β, and LBR[5,25,30,38]. Emerin was

**Fig. 1 | Reporter cell system reliably monitors intranuclear position of the *MyoD1* locus in vivo. a** Schematic overview of experimental strategy for monitoring the 3D nuclear position of the *MyoD1* locus in living myoblasts by inserting a 64x LacO array downstream of *MyoD1* coupled with expression of GFP-LacR. **b** Representative 3D reconstructed z stack of images showing intranuclear 3D position of the *MyoD1* locus in proliferating myoblasts (green dots). Scale bars are indicated on images. See also Supplementary Movie 1. **c** Immunofluorescence microscopy of proliferating and 4-days differentiated *MyoD1* and *Pax7* reporter myoblasts. *MyoD1* and *Pax7* loci are indicated by yellow arrowheads. Lamin A/C staining (red) marks nuclear border. Scale bars, 10 µm. **d** Schematic drawing depicting the analysis of the radial distance of gene loci to the nuclear periphery and normalization based on nuclear height. **e** Density plots of the normalized radial distance of loci to the periphery. Left: Distribution of *MyoD1* in proliferating (red) and differentiating (day 4, green) C2C12 myoblasts. $n_{prolif} = 106$, $n_{diff} = 120$, $p = 1 \times 10^{-4}$. Right: Distribution of *MyoD1* and *Pax7* loci in proliferating myoblasts.

$n_{MyoD1} = 208$, $n_{Pax7} = 79$, $p = 6.3 \times 10^{-4}$. $p$ values for loci were calculated using two-sample, two-sided KS test and corrected for multiple testing using Hochberg method. Numeric values displayed on violin plots represent median values. **f** Normalized radial position distribution of data shown in (**e**) displayed in violin plots. H3K9me2-marked region (gray-shaded horizontal bar) represents peripheral heterochromatin layer as measured in (**g**). ****$p = 1 \times 10^{-4}$, ***$p = 6.3 \times 10^{-4}$ $p$ values were calculated using two-sample, two-tailed KS test and corrected for multiple testing using Hochberg method. **g** H3K9me2 immunofluorescence staining of *MyoD1* C2C12 reporter myoblasts. Width of H3K9me2-marked layer is measured in rendered images (middle panel, yellow dots and lines illustrate measurement method for heterochromatin layer), bar graph depicts normalized mean thickness ± SEM of heterochromatin layer measured at 100 different locations. Scale bars, 10 µm. Images in (**c**) were processed with Fiji and in (**b**) and (**g**) with Imaris. Source data are provided as a Source Data file.

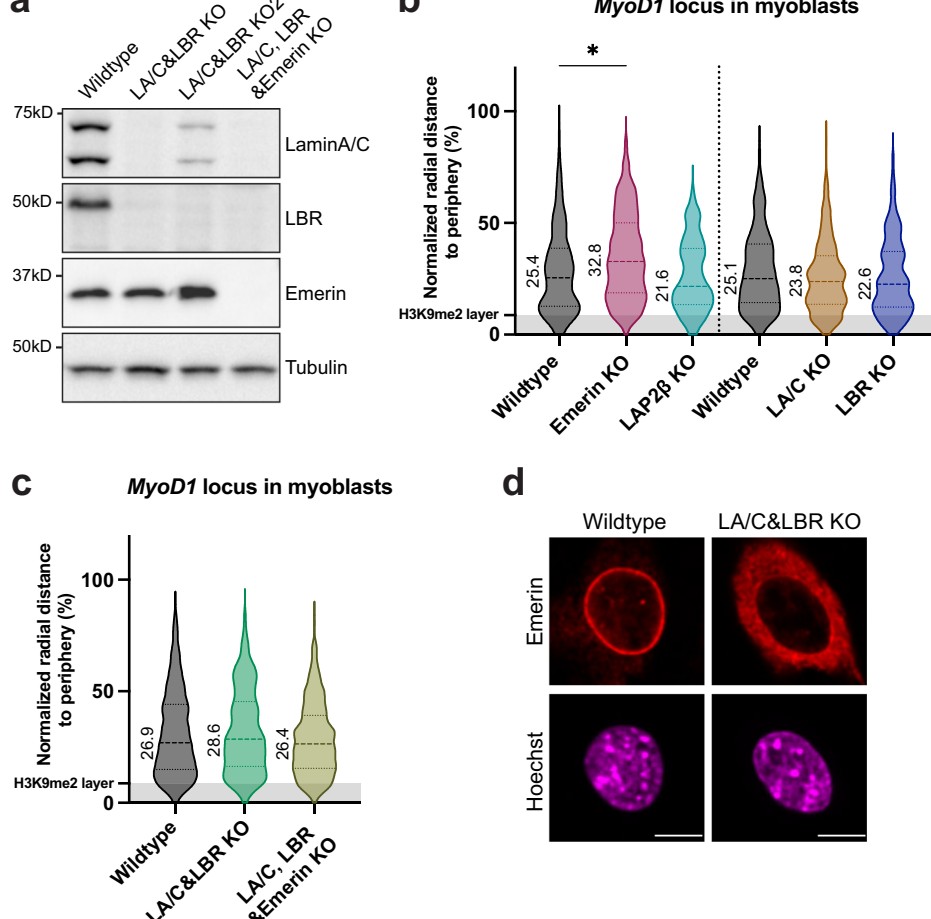

**Fig. 2 | Knockout of NETs involved in anchoring heterochromatin at the nuclear envelope does not affect the peripheral localization of *MyoD1*.**
**a** Immunoblot analysis of total cell extracts of proliferating wildtype and knockout C2C12 clones using antibodies to the indicated antigens. Normalized radial distance of *MyoD1* to the nuclear border in wildtype C2C12 myoblasts versus cells with (**b**) single knockouts, and (**c**) double or triple knockouts of indicated proteins. H3K9me2-marked heterochromatin layer is shown as gray-shaded bar. Dotted line indicates that experiments were performed separately using slightly different laser

settings. $n_{Wt} = 208$, $n_{emerin\ KO} = 333$, $n_{LAP2\beta\ KO} = 140$; $n_{Wt} = 141$, $n_{LA/C\ KO} = 403$, $n_{LBR\ KO} = 283$; $n_{Wt} = 722$, $n_{LA/C\&LBR\ KO} = 453$, $n_{LA/C,\ LBR\&Emd\ KO} = 300$. *$p_{Emd\ KO} = 0.02$, remaining $p$ values are non-significant ($p > 0.05$; two-sample, two-tailed KS test, Hochberg correction for multiple testing). Numeric values displayed on violin plots represent median values. LA/C lamin A/C, Emd emerin. **d** Localization of emerin assessed by immunofluorescence microscopy in control and lamin A/C-LBR double knockout myoblasts. Scale bars, 10 µm. Images processed with Fiji. Source data are provided as a Source Data file.

also previously identified as a potential peripheral tether of the *MyoD1* gene[53].

We depleted these genes individually, and in combination, with CRISPR/Cas9-mediated knockout, and confirmed the loss of proteins by Western blot analysis and/or immunofluorescence microscopy

(Fig. 2a and Supplementary Fig. 3a, b). Single-cell clones with single or combined gene knockouts were selected for analyzing the position of the *MyoD1* locus compared to wildtype controls. Single deletions of these proteins did not significantly alter the radial position of the *MyoD1* locus compared to wildtype cells, except for emerin-depleted

myoblasts, which displayed a subtle increase in the radial distance of *MyoD1* from the nuclear periphery (Fig. 2b). In control experiments we showed that protein depletion did not affect nuclear height (Supplementary Fig. 3c). Analyses of absolute, rather than normalized radial distance values also revealed no statistically significant change in *MyoD1* localization in knockout versus wildtype cells, except for emerin knockout myoblasts showing a subtle change in the positioning of the *MyoD1* locus (Supplementary Fig. 3d). However, the change in the radial position of *MyoD1* was very subtle in emerin-depleted cells and was not detectable in the lamin A/C, LBR and emerin triple knockout cells (Fig. 2c).

Remarkably, in cells lacking both, lamin A/C and LBR, which are known for their essential contribution to peripheral heterochromatin attachment[25], the *MyoD1* radial distance from the periphery was unchanged compared to wildtype cells (Fig. 2c). While the majority of emerin was relocated to the endoplasmic reticulum (ER) in the lamin A/C-LBR double knockout cells, a small fraction may remain in the inner nuclear membrane (Fig. 2d). Therefore, we also generated a triple knockout cell line, lacking lamin A/C, LBR and emerin (Fig. 2a). Importantly, *MyoD1* did not translocate to a more internal nuclear position in the triple knockout cells, suggesting that lamin A/C, LBR and emerin do not mediate the peripheral positioning of *MyoD1* (Fig. 2c and Supplementary Fig. 3d).

To confirm changes in heterochromatin organization in lamin A/C-LBR double knockout cells, we conducted immunofluorescence analysis of the heterochromatic histone mark H3K9me3 (Fig. 3a). H3K9me3 staining was substantially reorganized in the absence of lamin A/C and LBR, resulting in a reduction of H3K9me3-positive structures at the nuclear periphery and the formation of larger, but fewer heterochromatic foci in the nuclear interior compared to wildtype cells (Fig. 3b), as previously described[25]. Moreover, the nucleoplasmic to peripheral signal intensity ratio of the H3K9me3 signal was significantly increased in double knockout versus wildtype cells, confirming the release of heterochromatin from the nuclear periphery upon depletion of lamin A/C and LBR (Fig. 3c).

To test whether altered heterochromatin organization also affected the attachment of LADs to the nuclear envelope, we performed lamin B1 ChIP-qPCR analyses. While lamin B1 protein levels and localization remained unchanged (Fig. 3d and Supplementary Fig. 3e), the attachment of three different LAD regions to the nuclear periphery was abolished upon lamin A/C and LBR depletion, as seen by lamin B1 ChIP (Fig. 3e). Consistent with the ChIP-qPCR data, FISH analysis using a fluorescently labeled BAC probe within a LAD region confirmed the loss of peripheral association of the LAD (Fig. 3f, g). Despite the observed changes in spatial heterochromatin organization, total levels of the repressive histone marks H3K9me2, H3K9me3 and H3K27me3, were unchanged as seen in immunoblots (Supplementary Fig. 3f). We observed however, a subtle reduction in MyoD1 expression levels (Supplementary Fig. 3g), likely a consequence of the significant chromatin rearrangement caused by the depletion of lamin A/C and LBR[25].

Overall, these results show that the removal of heterochromatin, including LADs, from the nuclear periphery does not affect the peripheral localization of *MyoD1*, and demonstrate that the spatial compartmentalization of eu- and heterochromatin within the nucleus is not the primary determinant of peripheral *MyoD1* localization. Instead, these findings point to an active anchoring mechanism of *MyoD1* at the nuclear periphery, probably involving specific inner nuclear membrane proteins that are different from those attaching heterochromatic LADs.

### Identification of nuclear envelope anchors for *MyoD1*

In order to identify potential tethers of *MyoD1* at the nuclear periphery, we knocked out several candidate proteins of the inner nuclear membrane by setting up a CRISPR/Cas9-based experimental system capable of easily targeting a relatively large number of proteins, either individually or in various combinations. Cas9 endonuclease was stably expressed in the *MyoD1*-LacO reporter cell line, and synthetic sgRNAs designed to target the gene(s) of interest were transiently transfected (Supplementary Fig. 4a-g). Optimal knockout efficiency was achieved by simultaneous transfection of three sgRNAs targeting distinct genomic regions of the same candidate gene. Genome editing and knockout efficiency were assessed using the TIDE software (Tracking of Indels by Decomposition[55]) and immunoblotting or RT-qPCR analyses, as exemplified for the LEM2 knockout (Supplementary Fig. 4c-g). Depletion of the ubiquitously expressed NETs, LEM2, LAP1, and SUN1, did not change the radial position of *MyoD1* compared to wildtype controls (Supplementary Fig. 4f-h).

Next, we targeted a group of NETs expressed in skeletal muscle, which was reported to mediate nuclear envelope attachment of whole chromosomes (NET39, Tmem8a, WFS1, Tmem214)[40]. We sequentially knocked out these NETs generating a quadruple knockout cell line (Fig. 4a, Supplementary Fig. 5a) and measured the radial position of *MyoD1* in the different combined knockouts, as well as in single knockout cells. Several sequential knockouts displayed a significant change in the spatial positioning of *MyoD1* (Fig. 4b, left panel). While NET39 depletion alone caused only a subtle shift of the *MyoD1* locus towards a more central position within the nucleus, additional depletion of Tmem38a induced a statistically highly significant change in the localization of *MyoD1* towards the nuclear interior, which was further enhanced upon deletion of WFS1 (Fig. 4b, left panel). Additional knockout of Tmem214 did not lead to further release of *MyoD1* from the periphery, suggesting that primarily Tmem38 and WFS1 were involved in tethering *MyoD1* to the nuclear envelope. Single knockouts of Tmem38a and WFS1 also caused significant relocalization of the *MyoD1* locus towards the nuclear interior (Fig. 4b, right panel). Changes in the radial position of the locus were not due to nuclear height alterations in the knockouts, and measuring absolute rather than normalized distances yielded similar results (Supplementary Fig. 5b, c). Thus, we concluded that Tmem38 and WFS1 can independently tether *MyoD1* to the nuclear periphery.

Relocalization of the *MyoD1* locus towards the nuclear interior following depletion of Tmem38a or WFS1 did not correlate with increased transcription of the locus (Supplementary Fig. 5d). To exclude the possibility that release of the *MyoD1* gene was a consequence of premature cell differentiation triggered by Tmem38a and WFS1 depletion, we assessed the differentiation state of these cells. Both the proliferating wildtype and Tmem38a and WFS1 knockout cells did not express the muscle differentiation marker myosin heavy-chain (MyHC), suggesting that cells did not undergo premature myogenic differentiation (Supplementary Fig. 5e, f).

Overall, we identified the NETs Tmem38a and WFS1 as tethers for the *MyoD1* locus in proliferating myoblasts.

### ER membrane protein WFS1 localizes in the inner nuclear membrane

Next, we aimed to investigate the *MyoD1*-tethering functions of Tmem38a and WFS1 in more detail. However, we were unable to identify reliable antibodies for Tmem38a and thus initially focused the functional analyses on WFS1. WFS1 (Wolframin) is known as an ER protein containing nine predicted transmembrane domains (Fig. 5a[56]) and is linked to the autosomal recessive neurodegenerative disease Wolfram syndrome (WS), characterized by early-onset diabetes, optic atrophy, and hearing loss[57,58]. Functionally, it has been shown to be involved in intracellular $Ca^{2+}$ homeostasis[59] and vesicle transport in pancreatic β-cells[60], but its specific molecular functions are mostly unknown.

We first tested the intracellular localization and abundance of WFS1 in proliferating wildtype and Tmem38a knockout cells by immunofluorescence analysis and immunoprecipitation followed by immunoblotting (IP-WB). WFS1 is expressed at detectable levels in proliferating cells, primarily localizes to the ER and its expression

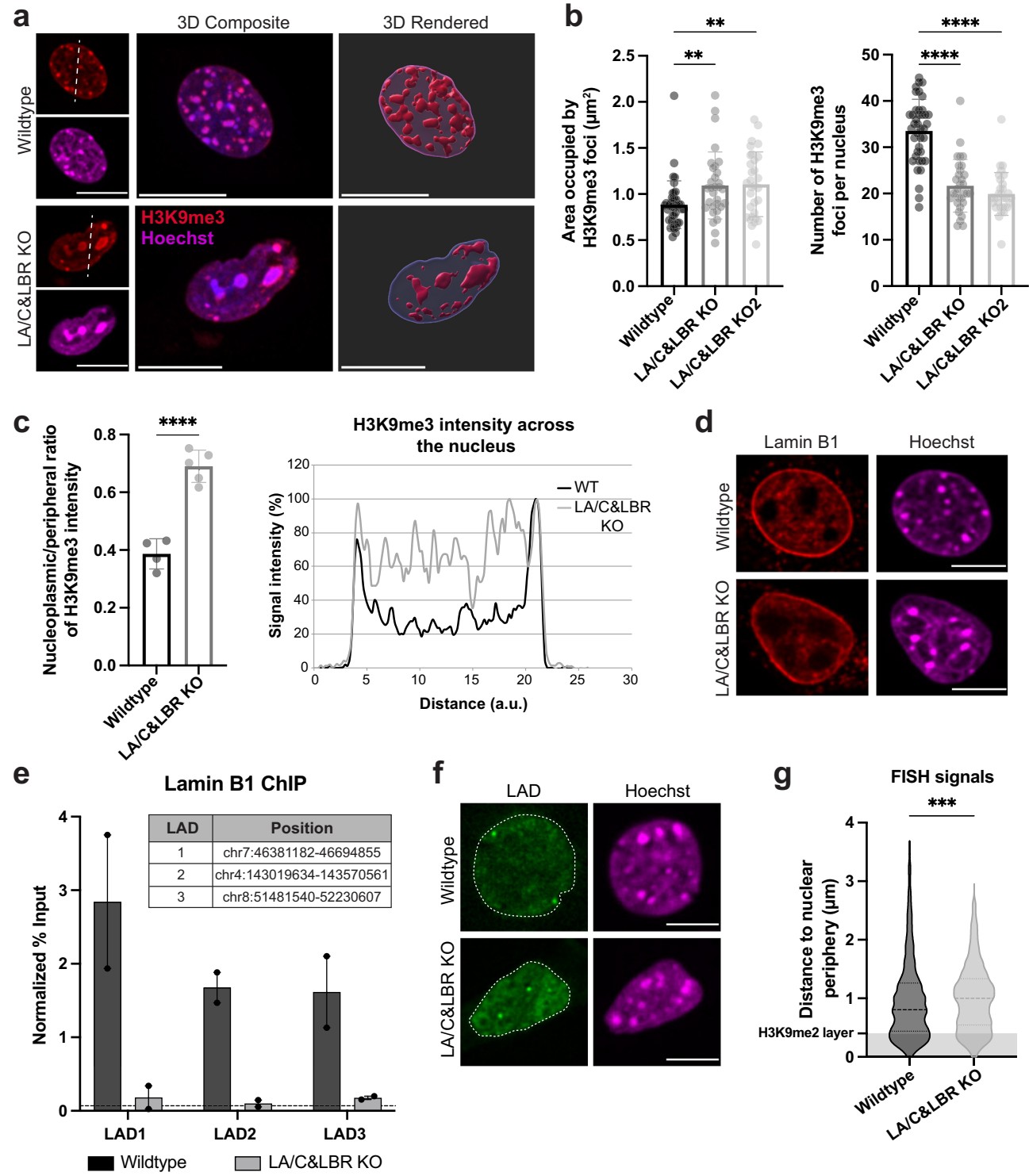

levels and cellular localization were unchanged in Tmem38a knockout myoblasts compared to wildtype cells (Fig. 5b, c). In immunoprecipitation-immunoblot analyses, WFS1 was detected as monomeric and partially dimeric protein as shown before[60] (Fig. 5c).

For testing WFS1 topology within the ER membrane, we imaged WFS1 knockout cells expressing ectopic WFS1 with a red fluorescence marker (mScarlet) inserted at its cytoplasmic N-terminus and an HA tag inserted at the ER lumenal C-terminus (Supplementary Fig. 6a-c). While the ER-lumenal HA tag was only detected upon Triton-X100 extraction, which permeabilizes all cellular membranes, and not upon Digitonin extraction, permeabilizing only the plasma membrane, the fluorescent

mScarlet signal and the cytoplasmic mScarlet antibody staining were detected upon both treatments (Fig. 5d). Detection of LAP2α, a nucleoplasmic protein, served as a positive control for the assay. Thus, the WFS1 N-terminus localizes to the cytoplasm and its C-terminus to the ER lumen.

In order to test whether WFS1 also localizes to the inner nuclear membrane, we expressed ectopic WFS1 with a FLAG Tag inserted at its nucleoplasmic/cytoplasmic N-terminus in WFS1 knockout cells and tested its correct localization and expression by immunoblotting and immunofluorescence analyses (Supplementary Fig. 6a, d, e). We then imaged these cells by high-resolution microscopy using antibodies to

**Fig. 3 | Depletion of lamin A/C and LBR induces reorganization of H3K9me3-marked heterochromatin and detachment of LADs from the nuclear periphery. a** Immunostaining and 3D reconstruction of H3K9me3-marked heterochromatin in wildtype and lamin A/C-LBR double knockout C2C12 cells. Scale bars, 10 µm. **b** Graphs show quantification of mean area per intranuclear H3K9me3-positive spot (left), and number of intranuclear foci per cell (right). Bar graphs represent mean ± SD. $n_{Wt} = 40$, $n_{KO} = 31$, $n_{KO2} = 31$ (single data points are displayed). Left graph: H(2) = 11.72, $**p_{KO} = 0.0096$, $**p_{KO2} = 0.0058$; right graph: H(2) = 52.07, $****p_{KO} = 9.8 \times 10^{-8}$, $****p_{KO2} = 6.6 \times 10^{-11}$ (Kruskal-Wallis test). **c** Nucleoplasmic over peripheral H3K9me3 signal ratio determined using line intensity profiles across the nucleus in wildtype and lamin A/C-LBR double knockout cells. Line graph on the right shows a representative H3K9me3 fluorescence intensity plot measured along the dashed lines in the images shown in (**a**). Bar graph displays mean values ± SD

$(n_{Wt} = 4, n_{KO} = 5)$. $t(7) = 8.31$, $****p = 7.1 \times 10^{-5}$ (two-tailed t-test). **d** Lamin B1 immunostaining of indicated cell lines. Scale bars, 10 µm. **e** Lamin B1 ChIP-qPCR analyses demonstrating loss of lamin B1 binding to LADs at the nuclear periphery of lamin A/C-LBR double knockout cells. Data represent mean values ± SEM of three biological replicates. Horizontal dashed line indicates the signal obtained using unspecific IgG antibodies in control ChIP. The genomic positions of tested LADs are indicated in the table. **f** Representative confocal images of fluorescence in-situ hybridization (FISH) signal using a GFP-labeled BAC probe for a LAD region (green dots). Dashed line indicates nuclear border. Scale bars, 10 µm. **g** Radial distance of LAD region to the nuclear periphery upon depletion of lamin A/C and LBR as detected by FISH. $n_{Wt} = 909$, $n_{KO} = 666$, $****p = 7.6 \times 10^{-4}$ (two-sample, two-tailed KS test). Images in (**a**) processed with Imaris, and in (**d**) and (**f**) with Fiji. Source data are provided as a Source Data file.

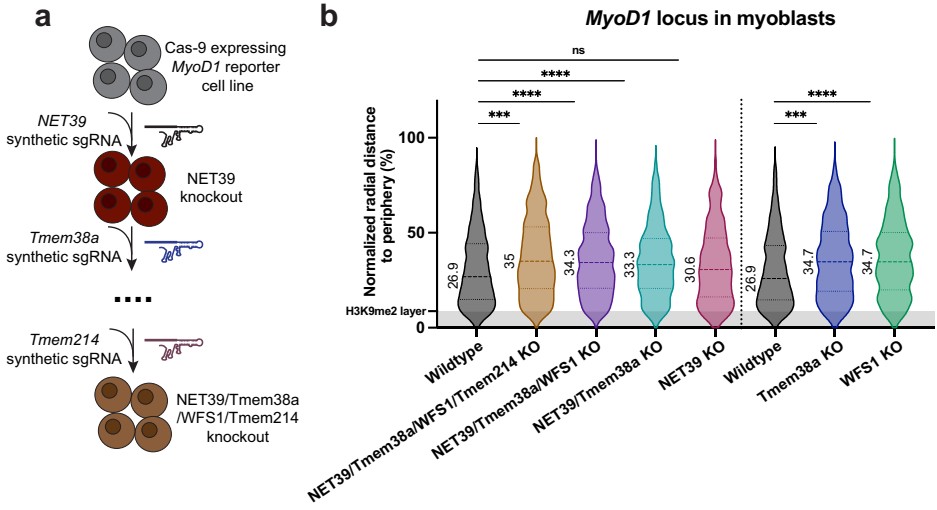

**Fig. 4 | Identification of nuclear envelope anchors for *MyoD1*. a** Schematic overview of the strategy for generating multiple protein knockout cell lines. *MyoD1* reporter C2C12 myoblasts constitutively expressing Cas9 were transfected with three synthetic sgRNAs targeting the gene of interest. Depletion of various candidate proteins was performed consecutively. **b** Detection of normalized radial distance of *MyoD1* to periphery in proliferating myoblasts following depletion of

indicated protein(s). Violin plots were generated from at least 450 data points. $n_{Wt} = 722$, $n_{4KO} = 454$, $n_{3KO} = 898$, $n_{2KO} = 685$, $n_{NET39\ KO} = 500$; $n_{Tmem38a\ KO} = 571$, $n_{WFS1\ KO} = 517$. $***p_{4KO} = 2 \times 10^{-4}$, $****p_{3KO} = 3.2 \times 10^{-6}$, $****p_{2KO} = 7.9 \times 10^{-5}$, $p_{NET39\ KO} = 0.6$; $***p_{Tmem38a\ KO} = 2 \times 10^{-4}$, $****p_{WFS1\ KO} = 1 \times 10^{-6}$ (two-sample two-tailed KS test, Hochberg correction for multiple testing). Numeric values displayed on the violin plot represent median values. Source data are provided as a Source Data file.

WFS1 and LAP2β or lamin A/C as markers for the inner nuclear membrane and the lamina underlying the nuclear membrane, respectively, and measured signal intensities across the nuclear border (Fig. 5e, upper panel). As depicted in the line plots and the insets in Fig. 5e (lower panel), the maximum signals of WFS1 and LAP2β coincide, while the lamin A/C signal maximum is slightly shifted towards the nucleoplasmic side relative to that of WFS1 (Fig. 5e, lower panel). These data are consistent with a localization of WFS1 in the inner nuclear membrane.

Altogether, our microscopic analyses showed that the transmembrane protein WFS1 mainly localizes throughout the ER in proliferating myoblasts, with the N-terminus facing the cytoplasm and the C-terminus in the ER lumen. High-resolution microscopy indicated that a fraction of WFS1 is also located in the inner nuclear membrane, overlapping with the inner nuclear membrane protein LAP2β, and may therefore interact with chromatin and/or genomic loci in the nucleus via its nucleoplasmic N-terminus.

### WFS1 associates with the core enhancer region of the *MyoD1* gene

To test whether WFS1 in the inner nuclear membrane can indeed associate with genomic regions containing the *MyoD1* locus, we performed WFS1 chromatin immunoprecipitation coupled with next-

generation sequencing (ChIP-seq). We applied a ChIP protocol using two-step crosslinking with Disuccinimidyl glutarate (DSG) and formaldehyde and sheared genomic DNA by 30 cycles of sonication and, following immunoprecipitation, 6 additional cycles of sonication (total 36 cycles). This ChIP protocol enriched for euchromatic (30 cycles) and heterochromatic (36 cycles) genomic regions[61] and allowed inclusion of both chromatin fractions in next-generation sequencing by generating chromatin fragments between 100 to 800 bp (Supplementary Fig. 7a-c). Furthermore, immunoblotting confirmed the presence of dimeric WFS1 protein in the immunoprecipitated samples (Supplementary Fig. 7d). Following sequencing, we identified WFS1-enriched genomic regions using the MACS peak caller at stringent condition settings (see Methods), which detected peaks with an average size of 478 bp, covering a total of 0.31% and 0.19% of the genome in the ChIP samples following 30 and 36 sonication cycles, respectively. In order to include all WFS1 peaks within both euchromatin- and heterochromatin-enriched genomic fractions in the subsequent analyses, we combined the peak sets derived from ChIPs following 30 and 36 sonication cycles. The combined WFS1 peak set covers 0.38% of the genome (Supplementary Fig. 7e). To confirm specificity of the peak calling approach, we analyzed the WFS1 ChIP versus input signal (RPKM) following 30 and 36 sonication cycles on merged peak sets, revealing a specific signal in the ChIP and only background signal in the

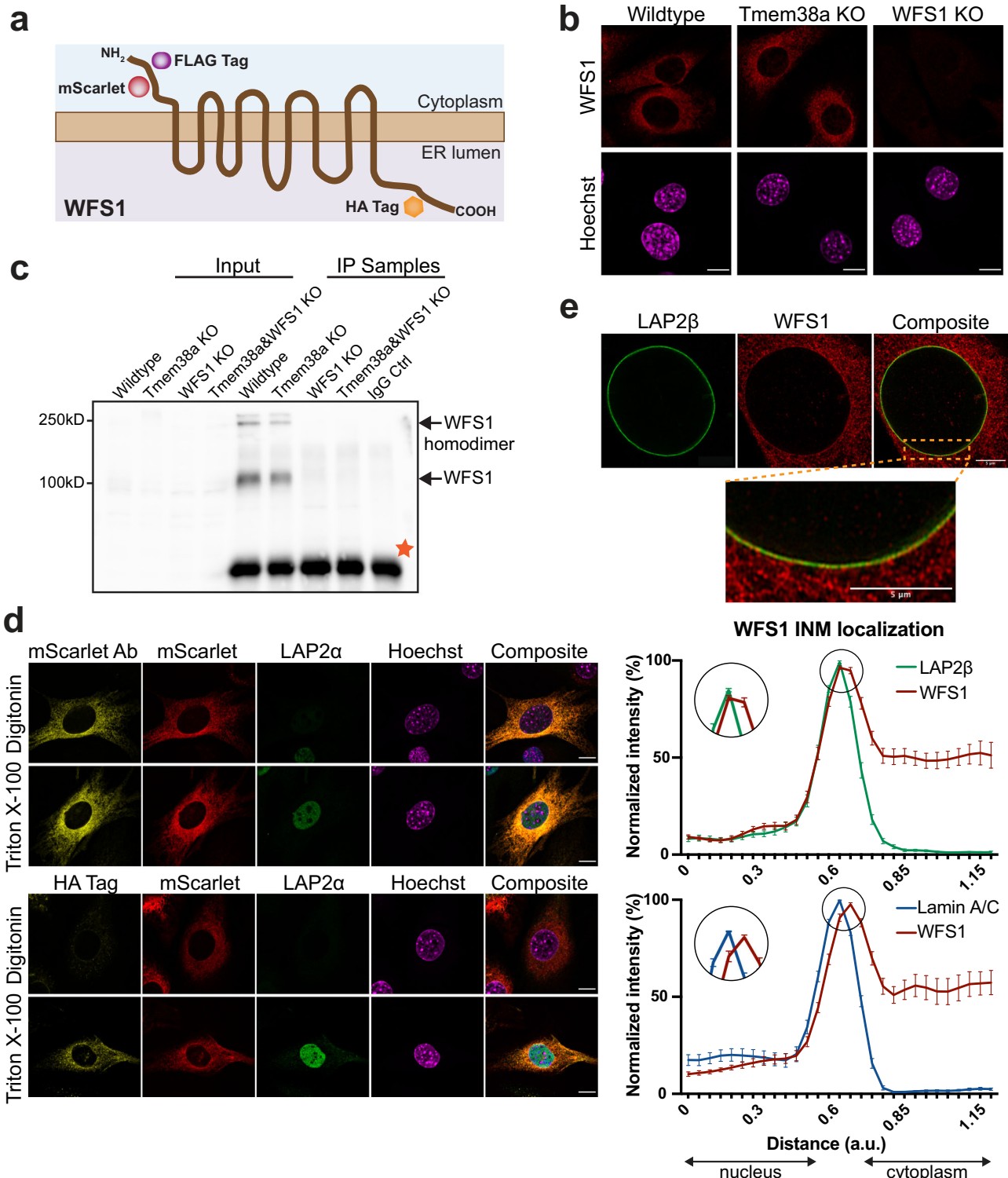

**Fig. 5 | WFS1 is a transmembrane protein of the ER partially located in the inner nuclear membrane. a** Schematic drawing of the predicted structure and orientation of WFS1 transmembrane protein. Positions of specific tags in ectopically expressed constructs are indicated. $NH_2$: N-terminus, COOH: C-terminus, ER: Endoplasmic Reticulum (**b**) Immunofluorescence analysis of WFS1 cellular localization in indicated genotypes. Scale bars, 10 μm. **c** WFS1 Western blots of total cell extracts (input) and of immunoprecipitated WFS1 (IP samples) using WFS1 antibody or IgG as control. Orange star marks unspecific band. **d** Representative confocal immunofluorescence images of WFS1 knockout cells ectopically expressing WFS1 tagged with FLAG, HA and mScarlet as indicated in (a), following membrane permeabilization with Triton X-100 (all membranes) or digitonin (plasma membrane only), using antibodies to the indicated antigens (anti-LAP2α

and anti-HA tag). mScarlet was either visualized by its own fluorescence (second panel) or by using an antibody (Ab) recognizing mScarlet (first panel). Scale bars, 10 μm. **e** WFS1 knockout cell clone ectopically expressing FLAG-tagged WFS1 was processed for super-resolution immunofluorescence microscopy using antibodies against WFS1 and LAP2β or lamin A/C. Representative super-resolution images for LAP2β- and WFS1- co-stained cells (AiryScan) are shown. Normalized signal intensity of WFS1 and LAP2β or lamin A/C was determined using line profiles across co-stained nuclei. Graphs display the relative localization of WFS1 to INM protein LAP2β (top) and intranuclear protein lamin A/C (bottom). Circles mark magnified areas of the graphs displayed on the left. Displayed are mean values ± SEM ($n_{top} = 25$, $n_{bottom} = 25$). Scale bars, 5 μm. Images processed in Fiji. Source data are provided as a Source Data file.

input sample (Supplementary Fig. 7f). Furthermore, in order to test the reproducibility of the WFS1 ChIP we also performed ChIP of ectopic FLAG-tagged WFS1 expressed in WFS1-depleted cells using the WFS1 antibody. The Pearson correlation coefficient between the endogenous versus ectopic FLAG Tag WFS1 ChIP signal was 0.81 in the 30-cycle replicate pair, and 0.79 in the 36 cycle replicate pair. MACS peaks called from FLAG Tag WFS1 ChIP samples were similar to those from WFS1 ChIP (Supplementary Fig. 7e), and heatmaps revealed a similar enrichment of WFS1 and FLAG Tag WFS1 signals (log2[ChIP/input]) on all peak sets (Supplementary Fig. 7g).

Having confirmed the specificity of the WFS1 ChIP and peak calling approach, we then analyzed the genomic region around the *MyoD1* gene in more detail. In the Integrated Genomics Viewer (IGV) browser, we detected enrichment of WFS1 at various genomic regions on chromosome 7 around the *MyoD1* locus (Fig. 6a). Regions significantly enriched for WFS1 around *MyoD1* were identified using the MACS peak caller. The nearest WFS1 peaks were located ~22 and 35 kb upstream of the *MyoD1* gene and situated within the *MyoD1* super-enhancer region, a region upstream of the *MyoD1* gene that contains several known regulatory elements[62,63] (Fig. 6a). Notably, the WFS1 peak located 22 kb upstream of *MyoD1* largely overlapped the core enhancer (CE) sequence (Fig. 6b), one of three cis-regulatory elements of *MyoD1*, alongside the distal regulatory region (DRR) and the proximal regulatory region (PRR). Epigenetic profile assessment of the *MyoD1* gene locus and its super-enhancer region in proliferating mouse myoblasts using publicly available ChIP-seq datasets for active and repressive histone marks and RNA Pol II revealed an accumulation of H3K27ac and H3K4me1 in the CE, while the promoter region (PRR) and gene body exhibited enrichment in H3K4me3 and Pol II, consistent with an active enhancer and gene promoter, respectively. Accordingly, repressive marks H3K27me3 and H3K9me3 were barely detectable in these genomic regions. The WFS1-bound region located ~35 kb upstream of *MyoD1* also exhibited an active enhancer-type epigenetic signature (Fig. 6a). In contrast, the DRR region was depleted of both, active and repressive marks. We confirmed significant enrichment of WFS1 at the active core enhancer sequence compared to the inactive distal regulatory region by ChIP-qPCR analyses (Supplementary Fig. 8a). In addition, we performed WFS1 ChIP-qPCR analyses in a WFS1 knockout single cell clone, revealing a significant reduction of the WFS1 signal at the active core enhancer sequence in WFS1 knockout versus wildtype cells (Supplementary Fig. 8b). Furthermore, WFS1 ChIP in Tmem38a knockout cells, in which the *MyoD1* locus is displaced from the nuclear periphery (see Fig. 4), showed that WFS1 enrichment on the active *MyoD1* enhancer was significantly reduced in Tmem38 knockout versus wildtype cells (Supplementary Fig. 8c). These results suggest that WFS1 in the inner nuclear membrane has no or reduced access to the active *MyoD1* locus when delocalized towards the nuclear interior in Tmem38 knockout cells.

When searching for common sequence motifs in WFS1-bound genomic regions using Homer DNA motif discovery algorithm, we found a highly significant enrichment ($p < 1\,e^{-150}$) of an enhancer box (E-box) DNA motif, associated with basic helix-loop-helix (bHLH) regulatory proteins (Fig. 6c). The identified E-box sequence (CAGCTG) was found twice within the CE sequence of *MyoD1* overlapping with the WFS1 peak.

Altogether, these results suggest that WFS1 anchors the *MyoD1* locus at the nuclear periphery. However, it does not associate with the gene directly, but primarily binds to its active, enhancer-type cis-regulatory elements at the nuclear periphery.

**WFS1 binds to active gene regulatory sequences genome-wide**
In order to test whether WFS1 may bind to other related genomic loci in proliferating myoblasts, we analyzed WFS1 enrichment on two genes previously reported to localize at the nuclear periphery, *Myf5*[53] and *Plxna2*[40]. IGV browser images of these genomic loci revealed WFS1

peaks at predicted enhancers of these genes, enriched in active histone marks (Supplementary Fig. 8d, e), and WFS1 ChIP-qPCR analysis confirmed enrichment of WFS1 on the *Plxna2* enhancer region compared to a control region downstream of the enhancer (Supplementary Fig. 8a). Thus, WFS1 associates with active enhancer regions of several peripherally located genes.

Next, we analyzed WFS1 enrichment on a genome-wide level using WFS1 MACS peaks distributed across all chromosomes. Although the nuclear periphery is enriched in heterochromatic genomic regions (LADs), only a minor fraction of WFS1-enriched regions overlapped with LADs (<5%), while the vast majority of WFS1 peaks was clearly distal to LADs (Supplementary Fig. 9a). In fact, GIGGLE analysis[64] revealed substantial similarities between WFS1 peaks and peak sets of active histone marks (H3K27ac, H3K4me2, H3K4me3) associated with promoter and enhancer regions, as well as accessible chromatin regions (ATAC-seq signal), suggesting that genomic regions bound by WFS1 reside in an active chromatin environment (Supplementary Fig. 9b).

In accordance with these findings, average signal profiles and heat map analyses show that the WFS1 ChIP signal was significantly enriched on accessible chromatin (ATAC-seq peaks) and on regions with active promoter and/or enhancer-associated histone marks (H3K27ac, H3K4me1, H3K4me3), while WFS1 was depleted in heterochromatic regions with the repressive H3K27me3 mark (Fig. 7a; histone ChIP-seq peak sets were derived from the CISTROME database). Additionally, average intensity signal plots and heat maps of these ATAC-seq and histone ChIP-seq signals around the center of WFS1 peaks show signal enrichments for the active histone marks H3K27Ac, H3K4me1, and H3K4me3, but not for the repressive histone marks H3K27me3 and H3K9me3 (Supplementary Fig. 9c). These data confirm the association of WFS1 with open, active chromatin regions genome-wide.

To further investigate the genome-wide association of WFS1 with enhancer and promoter-like sequences, we computationally compared the distribution of WFS1 binding sites with annotated candidate cis-regulatory elements (cCREs) in the mouse genome. Interestingly, ~55% of WFS1-bound sites mapped to regions with distal or proximal enhancer-like signatures (dELS or pELS; Fig. 7b), whereas randomized genomic regions with the same total length and similar distribution as WFS1 peaks showed only ~9% overlap with the enhancer-like sequences. Accordingly, the average WFS1 ChIP signal intensity on enhancer-like sequences was significantly higher than that on promoters or other genomic regions (Supplementary Fig. 9d). In addition, we generated an extensive C2C12 myoblast-specific enhancer map using the activity-by-contact (ABC) model[65], in which enhancer predictions are made based on the activity of an element (estimated by H3K27ac ChIP-seq and ATAC-seq), the 3D conformational contact with the gene promoter (determined by Hi-C), and gene expression data (determined by RNA-seq; Fig. 6a, "enhancers"). 10.05% of the predicted active enhancers using this algorithm overlapped with WFS1-enriched regions and showed a clear enrichment of the WFS1 signal compared to non-overlapping enhancers (Supplementary Fig. 9e). Genes predicted to be regulated by these WFS1-bound enhancers were enriched in GO terms related to actin filament organization and muscle development (Supplementary Fig. 9f). Moreover, transcriptional activity assessment indicated that 67.6% of genes linked to WFS1 overlapping enhancers were expressed in proliferating myoblasts (Fig. 7c), and epigenetic profiling of their predicted enhancers showed enrichment of active histone marks and high accessibility (Fig. 7d). Altogether, WFS1 associates primarily with active and accessible enhancers linked to expressed muscle-related genes on a genome-wide basis.

Overall, this study indicates that the peripheral localization of the *MyoD1* gene is independent of heterochromatin attachment to the nuclear envelope. Instead, we identify WFS1 as a specific anchor for *MyoD1* in the inner nuclear membrane of proliferating myoblasts, which associates with active regulatory elements of *MyoD1* and other

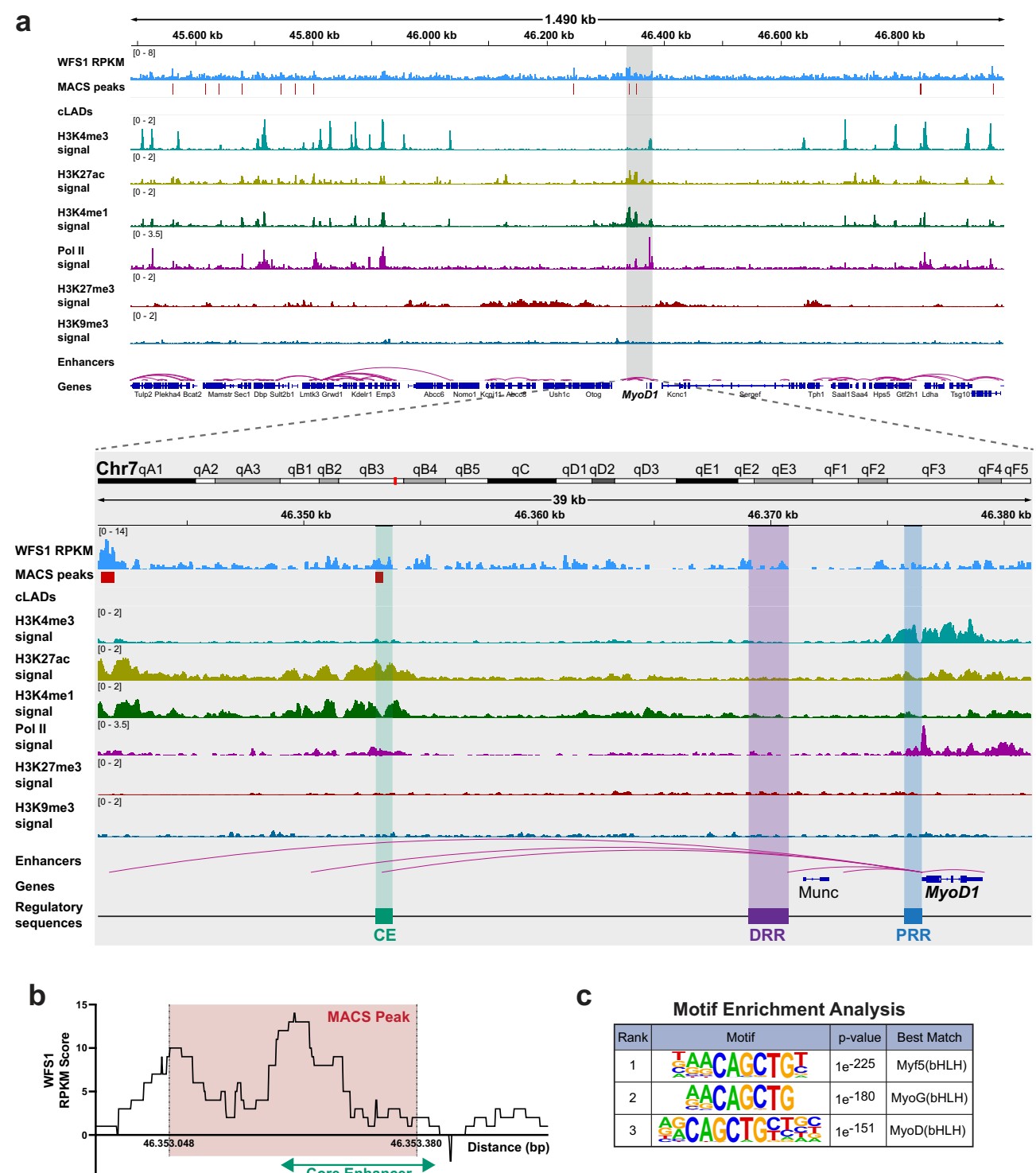

**Fig. 6 | WFS1 associates with the cis-regulatory core enhancer region of the *MyoD1* gene. a** Integrative Genomics Viewer (IGV) tracks of the mouse genomic region (mm10) containing *MyoD1* and its regulatory region (35 kb upstream) outlined by a gray box. Included are WFS1 ChIP tracks (RPKM, reads per kilobase per million) following sonication for 30 cycles, WFS1 peak regions identified by MACS, constitutive LAD regions (cLADs), ChIP-seq signal tracks of histone modification H3K4me3, H3K27ac, H3K4me1, H3K27me3 and H3K9me3, RNA polymerase II (Pol II) ChIP-seq signal track, enhancer predictions and gene annotations from NCBI reference sequence database (RefSeq) gene track. Known cis-regulatory regions of *MyoD1* are shown at the bottom (drawing not to scale). CE core enhancer, DRR distal regulatory region, PRR proximal regulatory region. **b** Graph showing reads per kilobase per million (RPKM) score of WFS1 ChIP signal within and flanking the core enhancer region of *MyoD1* (green line). Red-shaded area corresponds to coordinates of the closest WFS1 MACS peak. **c** Table showing relevant transcription factor binding sequence motifs identified in WFS1 ChIP-seq peaks by motif enrichment analysis using the HOMER algorithm for known and de novo motifs. *p* values are indicated. The common consensus, CAGCTG, is a canonical E-box sequence bound by basic helix-loop-helix (bHLH) transcription factors. Motif enrichment p-values were calculated using cumulative binomial distribution (sampling with replacement).

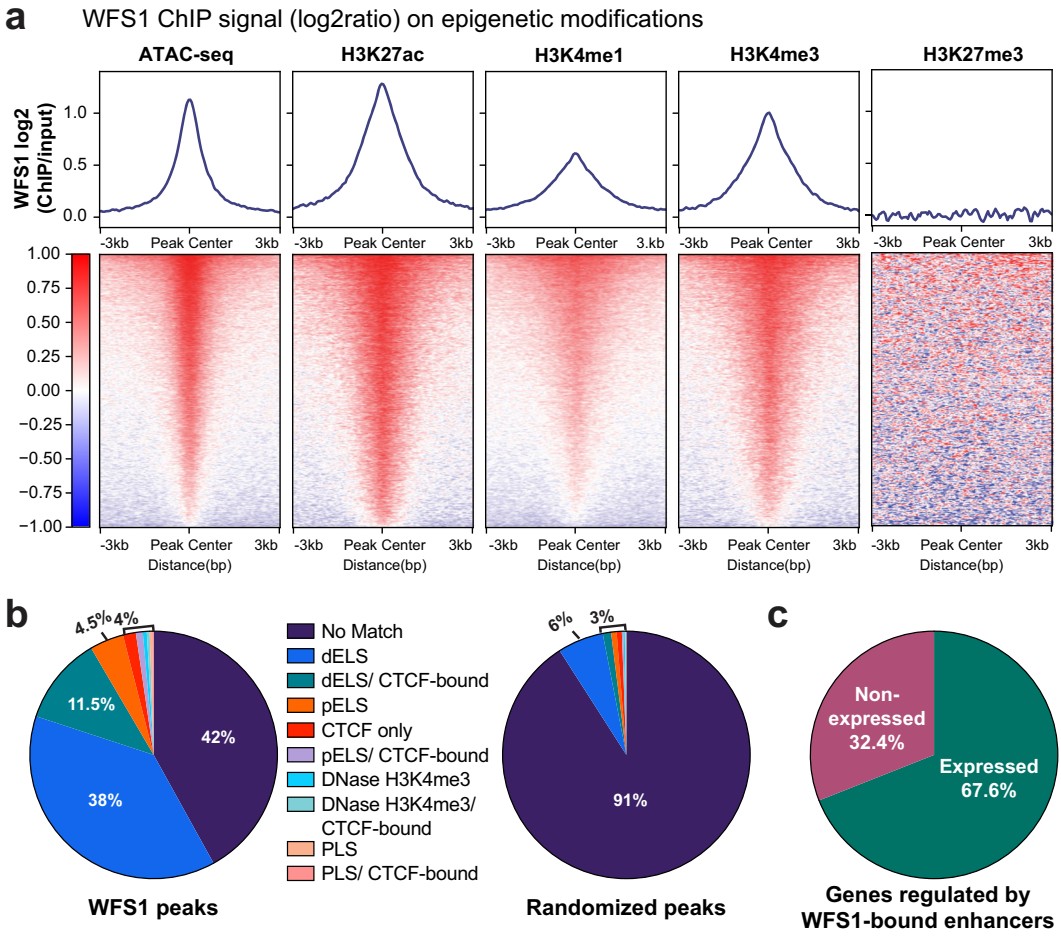

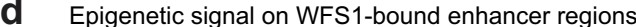

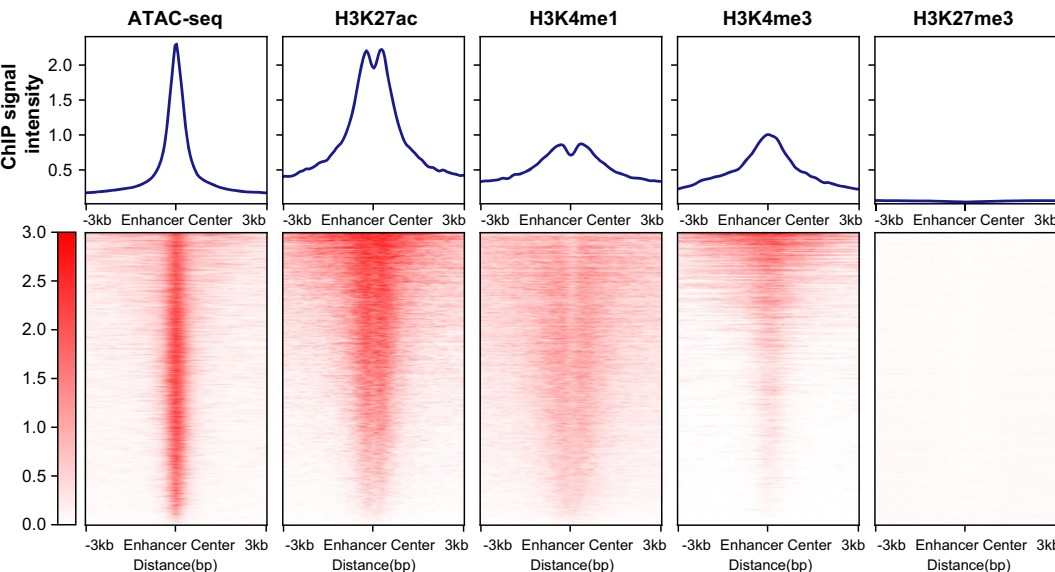

**Fig. 7 | WFS1 binds to open, accessible, and active gene regulatory sequences genome-wide. a** Average binding profiles and heat map showing enrichment of WFS1 (log2[ChIP/input]) (30 sonication cycles) on ATAC-seq, H3K27ac, H3K4me1, H3K4me3 and H3K27me3 ChIP-seq peaks ±3 kb from center. Heat maps are ranked according to WFS1 enrichment in descending order. **b** Degree of overlap (%) of WFS1 MACS peaks with candidate cis-regulatory elements from ENCODE (left). Overlap with randomized peaks is shown as control (right). dELS distal Enhancer-Like Sequences, pELS proximal Enhancer-Like Sequences, PLS Promoter-Like

Sequences. **c** Classification of genes associated with predicted WFS1-bound enhancers based on their expression status in mouse myoblasts. Minimum average expression threshold to classify a gene as expressed is 5 reads in at least two replicates. **d** Average binding profiles and heat maps displaying enrichment of ATAC-seq, H3K27ac, H3K4me1, H3K4me3, and H3K27me3 ChIP-seq signals on WFS1-bound enhancer regions ± 3 kb from center. Heat maps are ranked in descending order of ChIP-seq signal intensity.

expressed genes in the overall heterochromatic repressive environment of the nuclear periphery.

## Discussion

In mammalian cells, an association of genomic regions and genes with the nuclear periphery is commonly linked to heterochromatin formation and transcriptional repression. However, an increasing number of studies show that also actively transcribed genes can localize at the nuclear periphery, yet the underlying molecular mechanisms remain elusive. Here, we developed a myoblast reporter cell line and a semi-automated microscopic pipeline to efficiently map the 3D nuclear position of the *MyoD1* locus, which is transcriptionally active and localized at the nuclear periphery in proliferating myoblasts. Utilizing the generated reporter cell system, we investigate mechanisms and factors governing association of the *MyoD1* gene with the nuclear periphery. We show that localization of *MyoD1* is not dependent on the presence of heterochromatin at the nuclear periphery or mediated by the known heterochromatin tethers lamin A/C and LBR. Instead, we identify the nuclear envelope transmembrane protein WFS1 as a specific anchor for the *MyoD1* locus in proliferating myoblasts, interacting predominantly with active cis-regulatory elements of *MyoD1*. When analyzing WFS1 enrichment on a genome-wide level by WFS1 ChIP-seq we observed some degree of noise of the WFS1 signal. Despite this technical limitation of the WFS1 ChIP, which is in part due to the double crosslinking ChIP protocol required to identify WFS1 enrichment, we find WFS1 associated with a subset of active enhancers throughout the genome linked to expressed myogenic genes. Overall, these data suggest a mechanism involved in tethering regulatory elements of active genes to the nuclear periphery in myoblasts.

Compartmentalization of the genome into open euchromatin containing transcriptionally active genes, and transcriptionally repressive heterochromatin shapes genome organization[1,66]. Assembly and segregation of these compartments are mediated by phase separation, interactions with chromatin regulatory proteins, and association with nuclear structures, such as the peripheral lamina[16,22] or various nuclear bodies[13,67]. Spatial 3D genome organization often correlates with gene regulation[68]. Many silenced or moderately expressed genes localize at the nuclear periphery within the overall repressive heterochromatic environment, while most highly expressed genes localize in the interior integrated within the overall active environment. Repositioning of peripherally located repressed genes toward the nuclear interior usually coincides with their activation during developmental and differentiation processes[38–42]. The regulatory mechanisms of gene translocation from the periphery to the nuclear interior are not fully understood. One study showed that local chromatin decondensation of developmentally regulated peripheral loci was sufficient to induce their repositioning towards the nuclear interior[69], possibly by allowing de-mixing from the heterochromatic environment and association with open chromatin compartments. Thus, association of genomic loci with chromatin compartments may contribute to their 3D nuclear position. However, our study showed that deletion of the heterochromatic anchors lamin A/C and LBR at the nuclear envelope, causing translocation of heterochromatic H3K9me3-positive LADs towards the nuclear interior, did not affect the peripheral localization of *MyoD1*. This suggests that *MyoD1* is not tethered to the nuclear periphery indirectly, through association with the peripheral heterochromatin compartment, but likely by an active mechanism involving binding to nuclear envelope components.

A number of studies have demonstrated the "active" sequestration of developmentally regulated genes or loci at the periphery through specific tether proteins during processes such as myogenesis[40,53], cardiogenesis[38] and T cell differentiation[42,70]. Previously, a set of NETs (NET39, WFS1, Tmem38a, and Tmem214) was found to anchor genomic loci and entire chromosomes to the nuclear periphery during myogenesis in C2C12 cells[40]. In this study, we show

that WFS1, which was initially identified as an ER protein involved in the regulation of calcium homeostasis[56], serves as a tether for the *MyoD1* locus at the nuclear periphery in proliferating myoblasts. We demonstrate by high-resolution microscopy that WFS1 localizes partially also to the INM. Based on our observation that knockout of WFS1 leads to the release of *MyoD1* from the nuclear periphery, we hypothesize that the protein serves as an active peripheral anchor for the gene. Control experiments show that loss of WFS1 does not cause premature differentiation of myoblasts, which could induce *MyoD1* translocation indirectly. Thus, WFS1 may have at least two different functions: in the ER it affects calcium homeostasis and in the INM it serves as a specific tether for genomic loci. Proteins with dual functions have previously been described in the nuclear envelope, such as LBR and HDAC3, which fulfill roles in spatial organization of chromatin and cardiac genes independently of their enzymatic activity as a sterol reductase and deacetylase, respectively[26,38]. WFS1 is also abundantly expressed in the brain and pancreatic islets[71], where it is implicated in the regulation of intracellular $Ca^{2+}$ homeostasis[59] and ER stress response[60]. Although a tethering function of the protein in other non-muscle cell types cannot be entirely ruled out, WFS1 has so far been found to localize in the nuclear membrane only in muscle cells[40].

Notably, depletion of Tmem38a also resulted in the release of *MyoD1* from the nuclear periphery and a significant reduction of WFS1 binding to enhancer regions, suggesting the existence of multiple anchor proteins. Such additional tethers might introduce another layer of regulation for genes whose expression needs to be tightly controlled. Alternatively, Tmem38a might not interact with *MyoD1* and other loci directly, but affect their position indirectly by regulating WFSI. Further experiments will help to distinguish these two possibilities.

Our results raise the question about the physiological relevance of WFS1-mediated tethering of the core enhancer of *MyoD1* to the nuclear periphery. One can envisage several possible, non-mutually exclusive, scenarios: Firstly, the nuclear periphery is commonly considered a repressive environment, but recent reports highlight the presence of accessible, active chromatin regions within LADs, carrying active histone modifications, active or poised promoter and enhancer elements, and expressed genes, escaping the overall repressive environment[43,47,48]. As peripherally located *MyoD1* is expressed at basal levels in proliferating myoblasts and upregulated during differentiation[50], coinciding with its translocation to the nuclear interior, we hypothesize that WFS1-mediated tethering of the gene's active enhancer may physically inhibit enhancer-promoter interactions. Spatial segregation of the enhancer could ensure that transcription is solely driven by the promoter, thereby maintaining low expression levels of *MyoD1*. As we found genome-wide association of WFS1 with numerous distal and proximal active enhancers of expressed myogenic genes, WFS1 may regulate enhancer-promoter interactions of multiple genes, allowing their basal expression in proliferating myoblasts. Notably, aberrant changes in the expression of critical cell-fate genes have been linked to disturbances in developmental and differentiation processes in *C. elegans*[2], embryonic stem cells[38], and muscle cells[40,53]. Secondly, as a single gene can be regulated by the combined actions of several distal enhancers, and vice versa[72], it is crucial to ensure connections between the gene promoter and the appropriate enhancers at the right time. Spatial confinement of the core enhancer of *MyoD1* at the periphery by WFS1 may reduce the risk of unwanted, unspecific interactions with the promoters of genes other than *MyoD1* by limiting the ability of the regulatory element to dynamically explore its surroundings. Thirdly, expression of *MyoD1* in proliferating myoblasts and at different stages of differentiation is driven by distinct subunits of the TFIID transcription complex, whose spatial positioning in the nucleus mirrors the distribution of the gene[50]. Hence, it is tempting to speculate that positioning the enhancer and gene at the periphery may restrict their physical interaction with TFIID subunits in the nuclear interior and prevent premature upregulation.

While positioning of a gene within the nucleus can contribute to the regulation of its expression, it is not the only factor determining the level of transcription, as indicated by the fact that artificially altering intranuclear gene position does not always correlate with expression changes[40,44,45]. Similarly, in our study, release of the *MyoD1* locus from the nuclear periphery upon WFS1 depletion did not affect its expression. Thus, additional mechanisms likely involving epigenetic changes, tissue-specific transcription factors, chromatin remodelers, and DNA regulatory elements have to cooperate with gene positioning pathways to allow correct gene regulation.

In summary, we demonstrate here a pathway of spatial gene regulation involving tethering of active enhancers to the nuclear periphery, allowing at least a basal gene expression in an overall repressive heterochromatin environment and/or restricting premature upregulation of the gene prematurely. We postulate that enhancer-promoter interactions involved in fine-tuning gene expression in myoblasts are regulated, at least in part, by the WFS1-mediated association of enhancers with the nuclear periphery. Going forward, it will be of great interest to elucidate the functional relevance of peripheral enhancer tethering at the nuclear envelope and its specific role in the regulation of gene expression.

## Methods

### Cultivation and differentiation of myoblasts
C2C12 cells (Sigma Aldrich #91031101) were routinely maintained in a humidified incubator at 37 °C and 5% $CO_2$ in Dulbecco's modified Eagle's medium (DMEM, Sigma Aldrich #D6429) supplemented with 15% fetal calf serum (FCS, Sigma Aldrich #F7524, non-U.S. product), 2 mM glutamine (PAN-Biotech #P04-80100), 100 U/ml penicillin and 100 µg/ml streptomycin (P/S) (Sigma Aldrich #P0781). For reporter cell lines, the culture medium was additionally supplemented with 5 mM IPTG to repress binding of LacR to LacO repeats.

To induce differentiation, cells were plated on collagen-coated dishes (Collagen I, Rat Tail, Corning #354236) and differentiation medium (DMEM supplemented with 2% horse donor serum (Gibco #16050122), 2 mM glutamine, 100 U/ml penicillin and 100 µg/ml streptomycin) was added upon 80–90% confluency. The differentiation medium was replaced every 24 h for up to 10 days.

### Cell line generation
To generate *MyoD1* and *Pax7* reporter cell lines, C2C12 mouse myoblasts were transfected with the vector pSpCas9(BB)-2A-GFP (pX458, plasmid #48138, Addgene)[73] containing an sgRNA targeting a region 1 kb downstream of *MyoD1* or *Pax7*, as well as a repair template construct carrying the 64xLacO repeats flanked by regions homologous to *MyoD1* and *Pax7*, respectively (see also Supplementary Methods, chapter "Generation of repair template vector" and Supplementary Table 4 for primer sequences). Single-cell sorting of GFP-positive cells was performed using a BD FACS Melody™ Cell Sorter. Single-cell clones were cultivated and further characterized by genotyping PCR and long-range PCR (see also Supplementary Methods, chapter "genotyping PCR and long-range PCR" and Supplementary Table 4 for primer sequences).

Selected clones were transduced with pQCXIP-GFP-LacR vector (Addgene #59418), carrying the GFP-LacR coding sequence. GFP-positive cells were FACS sorted in bulk, selecting for low GFP levels to reduce background and facilitate microscopic detection of the loci.

To generate knockout clones, *MyoD1*-LacO cells expressing GFP-LacR were transfected with the vector pSpCas9(BB)−2A-mCherry (modified pSpCas9(BB)−2A-GFP[74]) carrying an sgRNA specific to the targeted gene. Single cell clones were FACS-sorted based on mCherry expression and gene knockouts were assessed using the TIDE software (https://tide.nki.nl/; see Supplementary Table 5, list of primers under section 'TIDE primers') and by Western blot. We designed 2 sgRNAs per gene (see Supplementary Methods, list of sgRNAs under section "Oligo sgRNAs" in Supplementary Table 6) and selected clones that showed full depletion of the targeted protein for further experiments.

All transfections were carried out using the Nucleofector 2b System in combination with the Cell Line Nucleofector® Kit V from Lonza, following the manufacturer's instructions.

The *MyoD1* reporter cell line stably expressing Cas9 was generated by transduction of *MyoD1*-LacO cells with the vector Lenti-CRISPR-V2 (plasmid #52961, Addgene[75]) carrying Cas9 from *S. pyogenes*, followed by selection using puromycin (1.5 µg/ml, Gibco #A1113803). To generate NET knockouts, *MyoD1*-LacO cells stably expressing Cas9 were transfected with small guide RNAs (sgRNAs, purchased from Dharmacon; see Supplementary Methods, list of sgRNAs under section "Dharmacon sgRNAs" in Supplementary Table 6) using Lipofectamine 2000 (Thermo Fisher Scientific #11668019) following the manufacturer's instructions. Briefly, 15,000 cells were plated on 24-well plates, sgRNAs were diluted in RNase-free Tris-HCl buffer pH 7.4 (Dharmacon #B-006000-100) and an oligomer-lipofectamine 2000 complex was prepared as follows: 2 µM total sgRNA (mix of 3 sgRNAs targeting the same gene) was diluted in 50 µl OPTIMEM serum reduced medium (Gibco #15392402), mixed with lipofectamine reagent, incubated for 20 min at room temperature and added to the cells in antibiotic-free medium for 48 h. Gene editing efficiency was assessed using TIDE (see Supplementary Table 5, list of primers under section 'TIDE primers').

All plasmids and cell lines generated in this study are available from the Lead Contact with a completed Materials Transfer Agreement.

### Immunoblotting and immunoprecipitation
To prepare cell lysates, cells were harvested in ice-cold phosphate-buffered saline (PBS, containing 1x Complete Protease Inhibitor cocktail from Sigma Aldrich #11697498001 and 1 mM EDTA from Invitrogen #15575020) and centrifuged at 250 $g$ (using a Heraeus Megafuge 2.0R) for 4 min at 4 °C. The collected pellet was resuspended in high-salt RIPA buffer (25 mM Tris-HCl pH 7.6, 500 mM NaCl, 1% NP-40, 1% Sodium Deoxycholate, 0.1% SDS, 2× Complete Protease Inhibitor cocktail, 1 mM EDTA, 1:100 Phosphatase inhibitor cocktail 2 and 3 from Sigma Aldrich), sonicated at 50% intensity for 3 seconds using a Sonopuls HD 200 sonicator from Bandelin and rotated for 30 min at 4 °C. Samples were then centrifuged at 9400 $g$ (using an Eppendorf Microcentrifuge 5420) for 10 min at 4 °C. Protein concentration was quantified using the Pierce BCA Protein Assay Kit (Thermo Fisher Scientific #23225) according to the manufacturer's instructions. After the addition of SDS PAGE sample buffer (186 mM Tris pH 6.8, 30% Glycerol, 6% SDS, 300 mM DTT, 0.1% Bromophenol blue), cell lysates were denatured for 5 min at 95 °C and proteins were separated on 10% SDS page gels using the MiniProtean electrophoresis system from Bio-Rad. Proteins were then transferred to a 0.2 µm Nitrocellulose Blotting Membrane (GE Healthcare #GE10600094, PAA) using the Mini Trans-blot cell (Bio-Rad). Membranes were incubated in 5% fat-free milk (Roth #T145.2) in 0.05% Tween/PBS (PBST) for 1 h at room temperature. Subsequently, the membranes were washed three times with PBST and incubated with primary antibody (see Supplementary Table 7, list of primary antibodies) diluted in 2% BSA/PBST containing 0.02% $NaN_3$ overnight at 4 °C. The next morning, membranes were washed three times with PBST, followed by incubation with the secondary antibody diluted in PBST for 1 h at room temperature. Membranes were washed three times with PBST and the specific antibody signal was visualized using SuperSignal West Pico Chemiluminescent Substrate (Thermo Fisher Scientific #34580) and the ChemiDoc Touch Imaging System by Bio-Rad.

For immunoprecipitation (IP), cells were harvested in ice-cold IP buffer (20 mM Tris-HCl pH 7.5, 150 mM NaCl, 2 mM EGTA, 2 mM MgCl2, 0.5% NP-40, 1 mM DTT, 1x Complete Protease Inhibitor cocktail and 1:100 Phosphatase inhibitor cocktail 2 and 3 from Sigma-Aldrich #P5726 and #P0044 respectively) and incubated for 10 minutes on ice.

Samples were centrifuged for 10 min at 2700 g at 4 °C (using a Heraeus Megafuge 2.0R) and the soluble fraction was used as input for the IP (0.7 mg protein/IP). IP samples were incubated with the desired antibody at a concentration of 5 µg/IP overnight at 4 °C, followed by incubation with protein A/G dynabeads (Pierce/Thermo Fisher Scientific #88802) for 4 h at 4 °C. Beads were washed three times in IP buffer and precipitated protein complexes were eluted from the beads using SDS PAGE sample buffer. IP samples were analyzed by immunoblotting.

## Immunofluorescence and digitonin treatment

Cells grown on glass coverslips (1.5 H from Marienfeld-Superior #0107032) were washed with PBS and fixed with 4% paraformaldehyde in PBS for 10 min at room temperature. Fixed cells were washed twice with PBS and incubated with 50 mM $NH_4Cl$ in PBS for 5 minutes at room temperature. Cells were washed twice with PBS and permeabilized with 0.1% Triton X-100 in PBS for 5 min at room temperature. Coverslips were then washed twice with PBS, incubated with primary antibodies (see Supplementary Table 7, list of primary antibodies) diluted in PBS at room temperature for 1 h and washed three times with PBS. After incubation with secondary antibodies diluted in PBS for 1 h at room temperature, coverslips were washed three times with PBS and mounted in VECTASHIELD Antifade Mounting Medium with DAPI (Vector Laboratories #H-1200-10). Imaging was performed using an inverse point scanning confocal microscope (Zeiss LSM 980) and 63× objective (Plan-Apochromat 63×/1.4 Oil). High-resolution images were obtained using an Airyscan 2 detector (32-channel GaAsP from Zeiss). Image processing was performed with FIJI or Imaris software as stated in Figure legends.

To permeabilize only the cell membrane, leaving internal membranes intact, cells were treated with 0.005% digitonin (Merck #300410) for 2 min at room temperature, instead of Triton X-100. The remaining steps were described for the immunofluorescence protocol.

## Measurement of H3K9me2 layer

Images of C2C12 myoblasts stained for H3K9me2 were analyzed using Imaris software. Following rendering of images, the boundaries of the peripheral H3K9me2 layer were labeled at opposite positions of the nucleus, and the distance between dots was measured (displayed as yellow dots and connecting dotted lines in Fig. 1g). The thickness of the H3K9me2 layer was calculated from an average of 100 distance values.

## Fluorescence in situ hybridization

C2C12 myoblasts were cultured on collagen-coated glass coverslips (1.5 H from Marienfeld-Superior #0107032) overnight, washed in PBS, and fixed in 4% paraformaldehyde in PBS for 10 min at room temperature. Cells were then washed twice in PBS for 5 min and fixation was quenched in 0.1 M $NH_4Cl$ in PBS for 10 minutes at room temperature. Cells were permeabilized in 0.5% Triton X-100 in PBS for 30 min, washed twice in PBS for 5 min and incubated with 20% glycerol in PBS at room temperature. Samples were transferred to 50% glycerol in PBS and stored at −20 °C overnight. Following a short calibration in 20% glycerol in PBS at room temperature, samples were subjected to five freeze-and-thaw cycles in liquid nitrogen. Samples were then washed twice in PBS for 5 min and treated with 0.1 M HCl for 30 min at room temperature to denature proteins. After washing twice in PBS for 5 min, cells were permeabilized again using 0.5% Triton X-100 in PBS for 30 min at room temperature. Samples were then equilibrated in 2x SCC for 5 min and in 50% formamide in 2x SCC for 30 min at room temperature. FISH probes (Empire Genomics; *MyoD1*−BAC clone RP24-35806, Genome position chr7: 46275014-46449836; LAD (chr8)− BAC clone RP23-10K23, Genome position chr8: 51369373-51570812) were diluted according to manufacturer's instructions. The inverted coverslips were then incubated with the probes on standard immunofluorescence glass slides and sealed with rubber cement (Fixogum

from Marabu #29010017000). Samples were heated to 78 °C for 2 min for simultaneous denaturation of the DNA and the probe and incubated for 48 h in a light-protected humid chamber at 37 °C. After removal of the rubber cement seal, samples were washed with the following buffers in this order: 2x SSC for 15 min at 45 °C, 0.2× SSC for 15 min at 63 °C, 2× SSC for 5 min at 45 °C and 2× SSC for 5 min at room temperature. Subsequently, samples were washed in PBS for 5 min at room temperature and incubated with Hoechst 33342 (1:10,000, Thermo Fisher Scientific #62249) for 10 min at room temperature to stain DNA. Finally, coverslips were washed in PBS for 5 min at room temperature and mounted in Vectashield mounting medium (Vector Laboratories #H-1000-10). Image acquisition and analysis were performed as described for the LacO reporter cell lines (see chapter 'Image acquisition and analysis' below).

## Chromatin immunoprecipitation

C2C12 cells were harvested using trypsin (Sigma-Aldrich #T2601) and the number of cells was determined using a CASY cell counter (OMNI Life Science). Cells were washed once in PBS, centrifuged for 8 min at 250 g (using a Heraeus Megafuge 2.0R) and fixed with methanol-free formaldehyde (Thermo Fisher Scientific #28906, 1% v/v final) for 10 min at room temperature. For WFS1 ChIP, double-crosslinking was performed. Pelleted cells were resuspended in 2 mM DSG in PBS (0.5 ml per 1 million cells) and incubated for 20 min with gentle rotation at room temperature, followed by incubation with 1% (v/v) methanol-free formaldehyde for 10 min at room temperature. Cross-linking was quenched for all ChIP samples by adding 2.5 M glycine to a final concentration of 125 mM and incubating for 5 min at room temperature. Cells were then centrifuged for 5 min at 400 g at 4 °C and washed twice with ice-cold PBS. Pellets were then resuspended in Wash1 buffer (10 mM HEPES pH 7.5, 10 mM EDTA pH 8.0, 0.5 mM EGTA pH 8.0, 0.25% Triton-X 100) at a concentration of 2 million cells/ml and kept on ice for 10 minutes. Samples were centrifuged for 5 min at 400 g at 4 °C and the pellets were resuspended in Wash2 buffer (200 mM NaCl, 1 mM EDTA pH 8.0, 0.5 mM EGTA pH 8.0, 10 mM HEPES pH 7.5) at a concentration of 2 million cells/ml, followed by another 5-min centrifugation at 400 g and 4 °C. Pellets were resuspended in lysis buffer (1% SDS, 10 mM EDTA pH 8.0, 50 mM Tris-HCl pH 8.1, 1x Protease Inhibitor cocktail, 0.1 mM PMSF) at a concentration of 1 million cells/100 µl and incubated overnight at 4 °C with gentle rotation. Samples were transferred to 15 ml sonication tubes containing 500 mg sonication beads (Bioruptor® Tubes & sonication beads, both from Diagenode #C01020031) and chromatin was sheared using the Bioruptor® Pico from Diagenode (30 cycles, 30 s on/30 s off per cycle). After sonication, chromatin was diluted in dilution buffer (16.7 mM Tris-HCl pH 8.1, 167.4 mM NaCl, 1.1% Triton X-100, 0.001% SDS, 1.2 mM EDTA pH 8.0) in a 1:0.5 ratio (chromatin:buffer). A small aliquot from each sample was kept to test shearing efficiency and the remaining samples were frozen in liquid nitrogen. DNA concentration was measured using a Qubit 4 Fluorometer and the Qubit broad range dsDNA Quantitation kit (both from Thermo Fisher Scientific #Q32850) according to manufacturer's instructions. To analyze the fragment size distribution of sheared chromatin samples, a small aliquot was analyzed on a microfluidic DNA chip using the Bioanalyzer 2100 (both from Agilent #5067-1504). This analysis revealed two DNA fragment peaks at 250 bp and 3 kb (see Supplementary Fig. 7b).

For chromatin immunoprecipitation, samples (3.5 million cells per ChIP) were diluted in dilution buffer supplemented with protease inhibitors to a final volume of 1.5 ml and the desired antibody was added to the respective sample (10 µg Lamin B1, Abcam #ab16048; 15.5 µg WFS1, Proteintech #26995-1-AP, RRID:AB_2880717; 15 µg IgG, Abcam #ab171870, lot #GR3409731-1). At this stage and prior to the antibody addition, 150 µl of chromatin were set aside as Input. Samples were gently rotated overnight at 4 °C, followed by incubation with protein A/G dynabeads (Pierce/Thermo Fisher Scientific #88802) for

4–5 h at 4 °C with gentle rotation (30 µl beads per sample). Beads containing immunoprecipitates were washed for 10 minutes at 4 °C while rotating with the following buffers in this order: 1x RIPA buffer (50 mM Tris-HCl pH 8.0, 150 mM NaCl, 0.1 % SDS, 0.5% Na-Deoxycholate, 1% NP-40), 1x High Salt buffer (50 mM Tris-HCl pH 8.0, 500 mM NaCl, 1% NP-40, 0.1% SDS), 1x LiCl buffer (50 mM Tris-HCl pH 8.0, 250 mM LiCl, 1% NP-40, 0.5% Na-Deoxycholate), and 2x TE buffer (10 mM Tris-HCl pH 8.0, 1 mM EDTA pH 8.0).

To elute chromatin, beads were resuspended in 200 µl elution buffer (2% SDS, 100 mM $NaHCO_3$, 10 mM DTT) and incubated for 30 min at room temperature in a shaker at 1200 rpm. The supernatant containing the chromatin was collected in a new 1.5 ml Eppendorf tube (#0030120086), and supernatants and inputs were decrosslinked using 4 M NaCl (10 µl per 200 µl sample), followed by incubation overnight at 65 °C in a shaker at 300 rpm. Samples were then supplemented with 0.5 M EDTA and 1 M Tris-HCl pH 6.5 (4 µl and 8 µl per 200 µl sample, respectively), treated with 10 mg/ml RNase (Thermo Fisher Scientific #R1253) (0.5 µl per 200 µl-sample) for 10 min at 37 °C in a shaker at 300 rpm, followed by Proteinase K treatment (final concentration 0.25 µg/µl, Thermo Fisher Scientific #EO0491) for 1 h at 55 °C in a shaker at 300 rpm. DNA was then isolated using the ChIP DNA Clean & Concentrator™ kit by Zymo Research #D5201 following the manufacturer's protocol and DNA concentration was determined using a Qubit 4 Fluorometer and the Qubit high sensitivity dsDNA Quantitation kit #Q32851.

Samples were then either analyzed by qPCR (see chapter "RNA isolation and qRT-PCR" below) or submitted to the Next Generation Sequencing (NGS) facility at the Vienna Biocenter Core Facilities (VBCF), which generated the library using the NEBNext® UltraTM II DNA Library Prep Kit for Illumina and sequenced the samples on an Illumina platform (NovaSeq SP) with SR100 XP mode (single-end reads; 100 bp length). For WFS1 ChIP-seq, half of the sample was submitted directly to the NGS facility, whereas the other half was transferred to 1.5 ml Bioruptor® Microtubes (Diagenode #C30010016) and sonicated for another 6 cycles (total 36 cycles) prior to NGS submission, to shear fragments larger than 800 bp and allow their subsequent analysis by NGS sequencing. Bioanalyzer DNA fragment size analysis of these samples revealed a broad DNA fragment peak at 250 bp (see Supplementary Fig. 7b).

Bioinformatic processing and analysis of sequencing data are described under "Computational methods".

## RNA isolation and qRT-PCR/qPCR

RNA was isolated using QIAGEN's RNeasy mini plus kit #74134 following the manufacturer's instructions, with RLT buffer containing DTT (20 µl of 2 M DTT/1 ml buffer). The concentration of the isolated RNA was determined using a DeNovix DS-11 spectrophotometer/fluorometer. RNA samples were stored at −80 °C. Removal of genomic DNA from RNA samples and cDNA synthesis were performed using the Thermo Fisher Scientific RevertAid First Strand cDNA Synthesis Kit #K1621 according to manufacturer's instructions. qRT-PCR was performed in triplicates for the genes of interest (using the appropriate primer pairs; see Supplementary Table 8, list of primers under section "qRT-PCR primers") and a housekeeping gene using KAPA SYBR Green 2x PCR master mix (Peqlab #KR0390) in an Eppendorf Realplex 2 Mastercycler.

For ChIP qPCR, eluted chromatin from immunoprecipitates and inputs were used undiluted and in a 1:100 dilution, respectively. qPCR was performed in triplicates for the regions of interest (using the appropriate primer pairs; see Supplementary Table 8, list of primers under section "ChIP-qPCR primers") using KAPA SYBR Green 2× PCR master mix (Peqlab #KR0390) in an Eppendorf Realplex 2 Mastercycler. Ct values obtained from immunoprecipitated samples were normalized to the respective input samples.

## Image acquisition and analysis

LacO reporter cells were maintained in IPTG-free medium 48 h before imaging to allow binding of GFP-LacR to LacO repeats. Cells were seeded on collagen-coated polymer bottom coverslips (Ibidi #80826) and stained with Hoechst33342 from Thermo Fisher Scientific #62249 (1:10,000 in phenol red-free DMEM medium from Gibco #21063029) for 10 min at 37 °C prior to imaging. Live cell imaging was performed using a Visitron Live Spinning disc/Nanodissection microscope equipped with a 63× objective (Plan Apochromat 63×/1.4 Oil) and an EM-CCD (1024 × 1024 pixel, 13 µm pixel size, 16 bit) camera at 37 °C and 5% $CO_2$. Two-channel multi-location z-stack images of samples were acquired.

Images were analyzed automatically using a custom-made Jython-script for FIJI/ImageJ. First, nuclear regions of interest (ROIs) were identified by intensity-based thresholding of the DAPI channel using the Huang2 algorithm. These ROIs were used to identify the fluorescent signal corresponding to the LacO-tagged locus of interest using the maximally projected GFP channel image. The absolute radial distance from the center of the fluorescent spot to the nuclear periphery was determined and normalized to the maximum radial distance per cell. In a 3D analysis, the Z dimension (height) is always the smallest distance, compared to X and Y. Therefore, the absolute values were normalized to the largest radial distance a locus can obtain in 3D, which is always Z/2. Finally, absolute values and normalized results were displayed in histograms and/or violin plots, where the nuclear border and nuclear center correspond to a normalized value of 0% and 100% respectively.

## Statistics and reproducibility

Data are presented as the mean ± SEM (standard error of the mean) or mean ± SD (standard deviation of the mean), as indicated in the figure legends. Statistical significance was estimated using two sample Kolmogorov-Smirnov test (KS-test), Kruskal-Wallis test followed by Dunn's test, two-tailed t-test, or Welch's t-test, as indicated in the figure legends, using GraphPad Prism (v10.0.2). The effect size was calculated using Cohen's d. When using KS-test, p values were corrected for multiple testing using the p.adjust function in R with the Hochberg method. Significance is represented as ns for not significant ($p > 0.05$), $*p < 0.05$, $**p < 0.01$, $***p < 0.001$, $****p < 0.0001$ ($1 \times 10^{-4}$). Immunofluorescence and Immunoblot experiments shown as representative examples in Figs. 1c, 2a, d, 3d, and 5b–d were repeated independently at least two times with similar results.

## Computational methods

The quality of raw ChIP-seq reads was evaluated using FastQC and reads were mapped to the Genome Reference Consortium GRCm38 (mm10) assembly using NextGenMap version 0.5.5 with default options. Reads with mapping quality score below 10 were filtered out using samtools view and the remaining reads were sorted and indexed. Peaks were called using MACS2 v2.1.4[76] with the settings *-q 0.05 -m 5 50 --keep-dup1* for 30 and 36 sonication cycles samples separately, and the two peak sets were combined using bedtools merge for all downstream analyses.

The reproducibility of the ChIP was checked by determining the Pearson correlation coefficient of the ChIP of endogenous WFS1 and of ectopic WFS1 expressed in WFS1-depleted cells. The mapping coverage of each sample was calculated in consecutive genomic bins of 2500 base pairs using deepTools multiBamSummary, and subsequently by using deepTools plotCorrelation with the settings --corMethod pearson --removeOutliers. The Pearson correlation coefficient between the 30 cycle replicate pair was 0.81, and between the 36 cycle replicate pair 0.79.

Motif analysis on WFS1 peak set was performed using Homer findMotifsGenome.pl function with settings -size 200 -len 8.

Overlapping peak coordinates between samples of interest were detected using bedtools intersect command.

The closest distances between genomic regions of interest were calculated using closest-features --dist from BEDOPS and illustrated in bins of distances.

A randomized peak set was generated using bedtools shuffle −chrom that randomly selects a genomic region for each peak on the same chromosome with the same length as the peak, and for randomizing regions only within ciLADs by providing the ciLAD coordinates using −incl setting.

ATACseq and histone modification ChIP-seq data originating from C2C12 myoblasts were downloaded from http://cistrome.org/db/ as BED peaks and as BigWig files, with the following IDs: ATACseq (GSM1972411), H3K27ac (GSM822514), H3K4me1 (GSM822512), H3K4me3 (GSM818946), H3K27me3 (GSM1117988), H3K9me3 (GSM1531008).

cLAD and ciLAD regions were downloaded from GEO with the ID GSE17051 for mm9 genome assembly, and their coordinates were converted to mm10 assembly coordinates using hgLiftOver from UCSC.

GIGGLE scores were estimated by comparing the WFS1 ChIP-seq peak set with all histone modification ChIP-seq peak sets available on the Cistrome database using Cistrome DB Toolkit[77]. Dot plots illustrate the assigned GIGGLE score, a measure of similarity between the WFS1 peak set and individual database-derived samples, suggesting samples may be binding to common intervals.

WFS1 log2ratio files were generated using deepTools bamCompare --scaleFactorsMethod SES, and visualized using the Integrative Genomics Viewer (IGV). Coverage tracks of ChIP reads were generated using bamCoverage from deepTools with RPKM normalization method. Heat maps and meta-plots of log2ratio and RPKM coverage signals were created using the computeMatrix and plotHeatmap functions of deepTools.

Coordinates of cCREs in the mouse genome were downloaded from the ENCODE Registry[78].

Average log2ratio signal values of WFS1 ChIP for regions of interest were calculated using bigWigAverageOverBed from kentUtils from UCSC, and the combination of dELS (distal Enhancer-Like Sequences) and pELS (proximal Enhancer-Like Sequences) from cCREs was used as the enhancer group. Regions 1 kb up and downstream of all mouse gene transcription start sites were used as the promoters, the combined genomic regions from GRCm38.102 ENSEMBL annotation were used as the genes, and genomic regions in between genes were used as the intergenic set.

Gene ontology enrichment was performed using the clusterProfiler v3.18.1 function enrichGO. Over-representation of gene sets was calculated compared to all mouse genes in the GRCm38.102 ENSEMBL annotation and results were illustrated using the dotplot function in R.

Enhancer prediction analysis was conducted utilizing the Activity by Contact Model of Enhancer-Gene Specificity[65] following the analysis pipeline described here: https://github.com/broadinstitute/ABC-Enhancer-Gene-Prediction. A genome-wide Hi-C interaction matrix was generated with Juicer v1.6 to be used in this pipeline, expression counts for each gene were generated with STAR v2.7.10b and Rsubread v2.8.2, and the sequence alignment maps of ATAC-seq and H3K27ac ChIP-seq were generated with NextGenMap v0.5.5, all using raw sequencing reads downloaded from SRA: Hi-C (SRR16220088), RNA-seq (SRX032210), ATAC-seq (SRR2999996) and H3K27ac ChIP-seq (SRX103212). Genes were categorized as expressed and non-expressed based on the expression counts using an arbitrary threshold of five average mapped reads in the two RNA-seq replicates.

## Reporting summary
Further information on research design is available in the Nature Portfolio Reporting Summary linked to this article.

## Data availability
All datasets generated during the current study are available in public repositories. The WFS1 ChIP-sequencing datasets generated in this study have been deposited to the Gene Expression Omnibus (GEO) repository with the accession code GSE253460. The complete results of the enhancer prediction analysis performed in the current study have been submitted to the Zenodo repository with the identifier 13270436. Source data consisting of raw images of the MyoD1 localization generated and analyzed in the present study are available on Zenodo with the identifier 13826126. The cLAD and ciLAD regions used in this study are available on GEO with accession code GSE17051. Previously published datasets downloaded from Cistrome Data Browser as processed signal tracks or peak sets, used for illustrating their signals on heatmaps and on genome browser figures, and for illustrating WFS1 signal on their peak regions are available in the GEO repository under accession codes GSM1972411 (ATAC-sequencing dataset), GSM822514 (H3K27ac ChIP-sequencing dataset), GSM822512, (H3K4me1 ChIP-sequencing dataset), GSM818946 (H3K4me3 ChIP-sequencing dataset), GSM1117988 (H3K27me3 ChIP-sequencing dataset), GSM1531008 (H3K9me3 ChIP-sequencing dataset). Previously published datasets used in enhancer prediction analysis, downloaded as raw sequencing data are available in the NCBI Sequence Read Archive (SRA) repository with accession codes SRR16220088 (Hi-C dataset), SRX032210 (RNA-sequencing dataset), SRR2999996 (ATAC-sequencing dataset), SRX103212 (H3K27ac ChIP-sequencing dataset). Source data are provided in this paper.

## Code availability
The code used for the 3D image analysis is available at: [https://doi.org/10.5281/zenodo.10785251].

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

## Acknowledgements

We thank Harald Herrmann, Institute of Neuropathy, University Hospital Erlangen, for providing the LBR antibody, Andrew S Belmont, University of Illinois for LacO repeat plasmids, and Eric Schirmer, University of Edinburgh, for valuable discussions on WFS1 and Tmem38a structure and function. Microscopic analyses and FACS sorting were performed by the Biooptics Facility at Max Perutz Labs. We especially thank Thomas Peterbauer for the valuable assistance in automating the acquisition of imaging data. Next Generation Sequencing was performed by the Vienna Biocenter Core Facilities. We are grateful to the CIBIV HPC cluster for the computational environment provided for the analysis of our genomic datasets. This research was funded in whole or in part by the Austrian Science Fund (FWF) [P29713-B28, P32512-B and P36503-B] to R.F. and a doctorate program funded by the Austrian Science Fund (FWF) [W1261-B28]. For open access purposes, the author has applied a CC BY public copyright license to any author-accepted manuscript version arising from this submission. K.G. and D.F. are recipients of a DOC Fellowship of the Austrian Academy of Sciences at the Max Perutz Labs, Medical University Vienna (ÖAW DOC 25725 and ÖAW DOC 25912, respectively); This work received funding from a Marie Jahoda fellowship of the University of Vienna to N.N., and from the European Union's Horizon 2020 research and innovation program under the Marie Skłodowska–Curie grant agreement no 754558 to C.K.

## Author contributions

K.G., R.F. and N.N. conceptualized the study; K.G. designed and conducted the experiments, performed data analysis and visualization; F.S. performed all bioinformatics analysis; T.N. developed knockout assay and cell lines; C.K. developed 3D image analysis method; D.F. provided technical support and performed some ChIP experiments; K.G., R.F. and N.N. wrote the manuscript with contributions from all co-authors; K.G., R.F. and N.N. acquired funding and provided supervision.

## Competing interests

The authors declare no competing interests.
