## [Transparent Peer Review file · Nature Communications]

MyoD1 localization at the nuclear periphery is mediated by association of WFS1 with active enhancers

Corresponding Author: Dr Roland Foisner

Version 0:

Reviewer comments:

Reviewer #1

(Remarks to the Author)

In 'Peripheral localization of the MyoD1 gene is independent of heterochromatin tethers and mediated by binding of WFS1 to active enhancers', Gerogiou and co-authors use a novel reporter system in C2C12 cells to track the position of the MyoD1 gene locus as they manipulate both heterochromatin and potential anchoring proteins, ultimately implicating the gene WFS1 as contributing to anchoring the locus to the periphery via binding of distal MyoD1 enhancer elements. Their data on WFS1 binding throughout the genome suggests a potential role for it in binding enhancers more broadly and link it to expressed genes in myoblasts.

Given the significant role that nuclear and genomic organization plays in the process of myogenesis – not to mention the central role that MyoD1 plays in myogenesis - their finding is of significant interest, and will likely be of interest not only to those who work in myogenesis, but to those who consider re-organization of gene loci in other cell types as well. WFS1 has previously been described as playing a role in nuclear organization in myogenesis, but at the level of larger genomic regions, while the authors here delve into a single (critical) genomic locus.

The manuscript overall is logically laid out, with careful attention to controlled experiments. The materials and methods section is particularly thorough, and seems to provide all necessary information for reproduction of experiments. However, I do have a number of questions/suggestions for improvement as listed below – a few where I appear to have a different interpretation from the authors of some of the presented data that I'm hoping can be reconciled (points #4, 6, and 7), and the majority looking for more explanation on either rationale or context/interpretation. I have listed my points to reflect the order in which data and discussion were presented in the manuscript, rather than grouping by major/minor or a similar schema:

1. The authors' approach of normalizing their radial distance analysis is predicated on correcting for maximal nuclear height, which the authors show to be essentially stable in their system in Suppl Fig 2. However, myocytes typically demonstrate elongation along at least one nuclear axis during differentiation (I believe that the authors would refer to this as the 'X' axis based on their model in Fig 1D).

Can the authors comment (or show data on their decision process) on the choice to normalize to 'Z', as opposed to the axis that appears to show the greatest change during differentiation – 'X'? (I will say that, given the data shown in Suppl Fig 2f and 2g of their non-normalized data, I think the authors have likely chosen a reasonable approach, but I think some additional explanation of the logic would be helpful.)

2. Suppl Fig 1f-h: there is mention that the images in f – the light microscopy data – was obtained on a light microscope at 100x magnification but that it seems like numerous nuclei and cells are present per panel. Is it supposed to be the data in 1g (the confocal images) that was obtained at 100x?

3. Figure 2 – is there a reason that Figure 2c (the Western blot) is referred to in text prior to the figure panels 2a and 2b? While I have no issue with the data in 2c, the flipped ordering made it somewhat confusing when reading.

4. Figure 2 – does the comment in line 216 on page 10 about 'applying a significance threshold of $p < 0.01$ ' mean that when you apply a threshold of < 0.05 , you *do* see a difference between wildtype and Emerin KO in regards to MyoD1 locus

positioning? Given that you also see a difference in the non-normalized distances plotted in Suppl Fig 3d, and that your other distance distributions show congruity between normalized and non-normalized distributions in both Figures 1 and 2, I would advocate that reporting a subtle increase in radial distance with Emerin KO would potentially be the more accurate way to describe this data (and I do not believe detracts significantly from the message of the manuscript).

5. For the various figures that include the normalized radial distance distributions (eg. 1f and g, 2a and b), I think it would be helpful to the reader if, in addition to the 'n' of values analyzed, the mean values were reported on the figures (perhaps above each violin in the plot?) I found myself often considering where the mean line is located on the plots and comparing between conditions, and a numeric value would have been of help.

6. For MyoD1 expression and the Histone ChIP shown in Suppl Fig 3 (panels e and g specifically), can the authors explain: 1) why there is so much more variability in the std deviation of expression of MyoD in wildtype cells compared to their knockouts? It seems that if that deviation was a bit lower in the wild-type samples that the authors would have seen statistically significant difference of expression (considering how small the standard deviation was between replicates in the KO cells) and 2) Why there is nearly a 3-fold difference in mean signal for H3K4me3 in the MyoD1 promoter between wild type and KO cells? I have a hard time reconciling that difference with the statement in line 232-233 on page 10 that the histone ChIP signals were similar between KO and wild type cells.

7. Related to the point above about H3K4me3 ChIP: the authors' statement in lines 253-254 on page 11 that their ChIP in LADs of H2K27me3 and H3K9me3 shows no difference between wild-type and KO cells seems surprising to me looking at the data in Suppl Fig 3g. Are those truly not different on statistical testing? The standard deviations look like they don't overlap, especially for H3K27. (I will note that I certainly agree with the authors' interpretation of the immunoblots showing no difference of overall histone mark levels in Suppl Fig 3f.)

8. Figure 3g: Can the authors expand on why the reported n for MyoD locus analysis is so very variable between conditions? This figure panel stands out in particular to me : the n is 600 – 900 per condition here, and some other conditions have far lower numbers (for instance, nLAP1 KO in Suppl Fig 4h is only 19).

9. Figure 5e: As the authors mention in their text, I appreciate a shift between peak intensities when comparing the top and bottom panels (LAP2beta/WFS1 and Lamin A/C/WFS1, respectively), but as someone who does not typically consider this sort of data, the shift appears rather subtle to me. Can the authors give either some sort of statistical comparison or additional context to the relative strength of the difference between panels?

10. Do the authors have a reference for their two-step crosslinking and increased sonication shearing approach or, alternatively, can they expand on the logic/evidence for that? I typically do not think about using that approach to ensure mapping signal to heterochromatic regions. And did the authors describe what size range their DNA was in after that extent of sonication? I didn't see comment on what size that resulted in.

11. The authors report WFS1 peaks numbering in the ~10000 - 20000 range across the genome in Suppl Fig 7, and report a high association (69%) between WFS1 bound enhancers and expressed genes in Fig 7c. Can they comment on whether they are suggesting those 69% of expressed genes in myoblasts are located at the nuclear periphery, as implied by the WFS1 binding? And if so, does that comport with any available data (either from their own work or other sources) on where expressed genes in myoblasts are located in the nucleus?

12. As mentioned by the authors, WFS1 has been previously implicated in mediating chromosomal position in myogenesis (Robson, et al. 2016) – can they specifically address whether that chromosome data helps shed light on their analysis of the MyoD1 locus (ie. is MyoD1 on one of the chromosomes analyzed specifically in that work?)

Reviewer #2

(Remarks to the Author)

This review is provided by Philippe Collas.

In this manuscript, Georgiou et al. report a novel mechanism of interaction of an expressed and differentiation-regulated gene locus to the nuclear envelope. This specifically involves, in C1C12 myoblasts, association of a distal (predicted) enhancer of the MyoD1 gene with WFS1, an ER transmembrane protein which the authors also provide evidence of localization in the INM. Combinations of locus tagging, differential detergent extractions, imaging data (live, immunofluorescence and FISH) including a new pipeline for semi-automated quantification of locus positioning in the 3D nucleus, Ca9-mediated KOs and ChIPseq analyses collectively allow the authors to identify and characterize WFS1 as a novel regulator of locus (enhancer) tethering to the nuclear envelope.

This is a well conducted and informative study combining imaging and -omics experiments supported by many controls. The work significantly adds to our understanding of gene regulation mechanisms at the nuclear envelope, a notoriously largely repressive environment. The manuscript is essentially divided into two sections: 1) identification and characterization of the localization of WFS1, from imaging and associated approaches, and 2) identification of the genomic role WFS1 may have in myoblasts.

The authors should be commended on the carefulness of the imaging experiments, where several complementary methods are used (reporter cell line for live imaging, extractions/IFs, FISH, and quantifications). These experiments are loaded with relevant controls.

I however have two main sets of comments, which should be address to strengthen the authors' claims (points 1-7), and allow them to start testing their interesting hypothesis (discussion) on WFS1 being a regulator of Myod1 expression during differentiation, by perhaps modulating P-E interactions (points 8-9).

1. Fig 5, S6 and related text: p15, the authors should attempt to tie together their interpretations of their differential TX/digitonin extractions on the orientation and ER/INM localization of WFS1, with the high-res microscopy shown in Fig 5 and S6, from which they suggest an INM localization based on a shift in the WFS1 vs lamin A/C immunofluorescence signal. Their claim on the INM localization of WFS1 would be strengthened this way.
2. Out of curiosity: why did the authors chose Pax7 as another locus to look at by imaging? What is know on Px7 anchoring to the nuclear envelope?
3. My major concern is however on the WFS1 ChIP-seq data, Fig. 6 and associated supp Figs. Several points follow on that, which should be addressed. From the claims made on the genomic enrichment of WFS1, one might expect punctual enriched sites detectable merely with accumulated normalized read counts, as for TFs or sharp histone peaks. Out of curiosity, what do such profiles look like or WFS1?
4. WFS1 is an integral protein with 9 tm domains. Is WFS1 actually solubilized and detected in the chromatin fraction used for ChIP? Is it detected in the IP (done under ChIP conditions)?
5. Several points here: The WFS1 ratio profiles are noisy, and MACS2 calls peaks, it seems, not as much from WFS1 signal strength but mainly from (low) surrounding signal strengths; in other words, the profiles reveal many more signals, specific or not, than what is detected by MACS2 (Fig. 6a,b). Given the log2 ratios shown, the ChIP-seq data themselves, and therefore peak calling, may not be extremely convincing. So:
 - a. Is another WFS1 Ab available to validate the Ab used I the study?
 - b. The authors should validate their seq-data with by ChIP-qPCR in C2C12 using primers tiling "enriched" and non-enriched sites around the Myod1 gene and predicted enhancer sites (referring to Fig 6b). Can the seq profiles be replicated?
 - c. How much of the "noise" could be due to the double cross-linking? I assume the authors have tried only formaldehyde, and this was unsuccessful – is that so?
6. It would be reassuring to see a ChIP-seq analysis of WFS1 in C2C12 cells KO for WFS1.
7. What is the chromosome by chromosome distribution of WFS1 ChIP-seq signals and peaks? Is WFS1 enriched (given the type of analysis done) on specific (more peripheral?) chromosomes? Related to this: if WFS1 is an INM protein, how would the authors explain a distribution on (all?) chromosomes?
8. Ideally, it would be good to see WFS1 ChIP-seq data under a condition where WFS1 is delocalized from the INM; if I am not mistaken however, this may be tricky since even a KO of Tmem38a does not delocalize WFS1 from the INM. However: since KO of Tmem38a delocalizes Myod1 from the nuclear periphery (their imaging data), it would be good to test by ChIP-qPCR of WFS1 if, in Tmem38a KO cells, the Myod1 gene and enhancer(s) are dissociated from WFS1. This would strengthen the role of WFS1 in anchoring Myod1.
9. To provide relevant additional epigenetics background around Myod1, it would be informative to establish the H3K9me3 profile of C2C12 cells, from public or own ChIP-seq data. In particular, the heterochromatin landscape in the Myod1 region upstream of Myod1 and therefore of the LAD reported in Fig 6a would be useful to see. H3K9me3 should be included in the analyses (Fig 7 and relevant supp Figs).
10. They authors raise the interesting hypothesis of WFS1 being a regulator of Myod1 expression and P-E interactions. Fully testing this idea would require doing HiC or the like in myoblasts and myotubes, which may be beyond the scope of this study. However, the idea should be addressed by a ChIP-seq analysis of WFS1 and relevant histone marks (H3K27ac at least; H3K9me3) in myotubes. Should HiC or other enhancer prediction data also be available for differentiated C2C12 cells (myotubes), a similar analysis of what the authors report in myoblasts should be carried out. Altogether, these experiment would enhance the significance of their findings.
11. The authors extend their findings beyond Myod1 by providing GSEA terms of expressed genes whose predicted enhancers are also, it seems, bound by WFS1. Providing 1-2 more examples, even only through a browser views, would be interesting.
12. Could the authors speculate on whether WFS1 would also be involved in such tethering functions in other cell types / lineages, based on known expression profiles?
13. Lastly, out of curiosity, could the authors try to speculatively unify the role(s) of WFS1 in the INM, both as a regulator of Ca homeostasis and as enhancer tether? It is interesting that several proteins of the INM, e.g. LBR, have enzymatic and anchoring functions.

Reviewer #3

(Remarks to the Author)

Georgiou and colleagues investigate the role of mechanisms underlying positioning of the MyoD1 locus in myoblasts. They use a combination of DNA FISH, immunofluorescence, and occupancy studies in a MyoD1 reporter line to describe the role of a nuclear envelope protein, WFS1, in mediating MyoD1 positioning. The authors suggest that WFS1, an ER membrane protein, and not heterochromatin-related mechanisms, is responsible for peripheral tethering of MyoD1. Overall, the manuscript is well-written and data is presented well. The mechanisms by which loci get tethered get to different nuclear landmarks (in this case the nuclear lamina) is of interest. However, several conceptual and technical flaws dampen my enthusiasm substantially (outlined below) for the current version.

1. The WFS1 occupancy data is very difficult to interpret. First, the signal/tracks shown in Fig 6a appear to be very noisy. The signal appears widespread and the logic behind MACS peaks identifying areas of local signal enrichment is not clear. Moreover, how the replicates compared to each other in terms of peaks and signal intensity across the genome, should be presented. Given no occupancy data has been previously been published for WFS1 – the data would be much more convincing if a negative control was included (WFS1 ChIP from WFS1 KO cells or WFS1 ChIP in conditions in which a

competitive peptide to the ab is introduced).

2. Related to the ChIP-seq data, while the MACs peaks show a potential enrichment in enhancers – what fraction of signal is found in enhancers? Based on the tracks, only a small amount would be, which makes claims regarding enhancers being enriched for WFS1 occupancy tenuous.

3. I would encourage the authors to show more of their FISH quantification data as absolute distance from the nuclear periphery/lamina as opposed to the percentage measure they use most often. The absolute distance provides better context for how far the locus is from the periphery as opposed to the current percent data. In addition, how the authors measured H3K9me2 thickness is not clear.

4. Based on the authors' current work, it would be helpful if they could nominate other loci that may be WFS1 responsive in terms of positioning? This would strengthen the impact of the paper, as currently it is unclear if MyoD1 is the only locus whose positioning is dependent on WFS1.

5. The WFS1 localization data is primarily based on exogenous expression. While it is well-done, it is still difficult to interpret because wouldn't most proteins which are over-expressed show some accumulation/trafficking in the ER? Have the authors considered endogenous ab staining and/or knocking a GFP tag into the WFS1 locus?

6. In the LMNA and LBR KO studies (Fig 3), the authors conclude that LADs move away from the periphery based on 3G. However, the probe is infrequently at the periphery in the WT cells based on the plots provided? Hence, perhaps it is expected that the positioning is not LMNA/LBR-dependent.

Version 1:

Reviewer comments:

Reviewer #2

(Remarks to the Author)

Georgiou et al. now provide a substantially revised and improved manuscript. All my comments have been addressed thoroughly with an impressive array of additional experiments, additional controls at various levels, and refined conclusions and interpretations. I have no further comment and find the manuscript acceptable for publication.

(Remarks on code availability)

Reviewer #3

(Remarks to the Author)

This is a revised manuscript by Georgiou and colleagues describing the role for WFS1 in mediating positioning of MyoD1 relative to the nuclear periphery (though it is unclear if WFS1 KD affects myogenesis). Reviewer 2 and myself both raised concerns about the WFS1-ChIP-seq datasets. While the ChIP-qPCR is supportive and I appreciate the additional technical data provided, I remain concerned about the quality, accuracy and interpretability of the WFS1 ChIP-seq. Suggestions to strengthen the analysis of the datasets:

1. While CRISPR did not work to KO the gene, the authors could consider siRNA to reduce levels and then perform ChIP-seq. While similar peaks were called at different thresholds (and this is supportive that the peak calling method is ok), it does not address the signal:noise characteristics of the chip-signal.

2. The authors should more stringently assess the individual biological replicates (each of the 30 cycle pair versus each of the 36 cycle pair versus the merge - a 5 way comparison).

a. The authors should call peaks on their individual biological replicates from the 30 and 36 cycles (not merged) using the same parameters used in the current manuscript and compare these to each other and the peaks from the current ones derived from the merged datasets. Per the methods, it appears 2 replicates per condition (30, 36 cycles) were generated and merged. Are the peaks the same across all the replicates compared to the merged?

b. The authors should compare the signal intensity specifically at called peaks in the merge to signal at the same location across all 4 replicates. These data can be plotted as a meta plot - merged and each individual replicate.

3. The individual replicate input tracks are not provided. If the authors called peaks on the input replicates using the same parameters used in the current manuscript- how often do the peaks overlap with the ones identified in the WFS1 ChIP-seq?

4. Metaplots of the merged and individual replicates should be provided showing signal at the same regions shown on Figure 7A.

5. The authors should provide a browser session with the raw and processed individual replicates and merged data displayed. The data from GEO is not available to be downloaded.

6. Fig 9E - the signal at “non-bound” enhancers should be compared to “bound regions” (and ideally permuted) - not random regions across the genome.

Other residual concerns:

7. The authors' manuscript title is “Peripheral localization of the Myod1 gene....”, but - as mentioned in my original review - the probe in Fig 3 is infrequently at the periphery. Thus, it is not relevant that occasionally LADs are not at the periphery. Moreover, given the title, it makes it hard for me to understand how lack of peripheral positioning makes it an applicable locus to support their title.

8. Please list the H3K9me2 ab used in the methods.

9. Are the cells cross linked in the experiments in Supp Fig 7D? It is unclear from the figure legend. Also, an input is not included.

10. SFig 3G (Left) and SFig 8D - the experiment should be repeated with biological replicates and not technical.

11. Fig 5C - is the WT:NET39 KO comparison statistically different?

(Remarks on code availability)

I do not have expertise to review code.

Reviewer #4

(Remarks to the Author)

I did not review the initial version of this manuscript, but have been asked to evaluate the authors' response to reviewer 1. After reading of the manuscript and the response, I find that the authors have diligently addressed the original points by reviewer 1. The same applies to their responses to the other reviewers. While there are some loose mechanistic ends, the reported findings point to a novel mechanism of gene positioning and will be of interest to many in the field.

(Remarks on code availability)

I am not qualified to review code.

Version 2:

Reviewer comments:

Reviewer #2

(Remarks to the Author)

-

(Remarks on code availability)

-

Reviewer #3

(Remarks to the Author)

I appreciate the authors efforts and additional experiments to address my concerns and the new data that is provided. Examining the data on the genome browser link provided by the authors suggests some degree of noise - at least in part likely driven by the double cross linking required to examine WFS1 enrichment. This is also consistent with the residual ChIP-qPCR signal observed in the KO cells. I suggest that the authors include this point as a technical limitation in the discussion.

(Remarks on code availability)

I am not qualified to review code.

We are grateful to the reviewers for their in general favorable evaluation and the insightful and constructive comments.

According to the reviewers' suggestions we performed several new experiments to demonstrate the specificity of the WFS1 ChIP and addressed all other comments of the reviewers. We have made substantial changes to the manuscript adding several new data and Figure panels to address all points. The main changes in the manuscript are:

- To address the specificity of the WFS1 ChIP we performed several new experiments.
 - 1) We confirmed the enrichment of WFS1 on several active enhancer regions of *MyoD1* and two additional peripheral genes (*Myf5*, *Plxna2*), located on 3 different chromosomes, versus non-active enhancers and/or genomic regions around these genes by ChIP-qPCR analyses (**new Supplementary Fig. 8a-c**).
 - 2) We performed WFS1 ChIP-qPCR experiments in *Tmem38a* KO cells, in which the *MyoD1* locus is detached from the nuclear envelope. Importantly, we find a significant reduction in WFS1 signals on active enhancers of *MyoD1* and another peripheral gene in *Tmem38a* knockout versus wildtype cells. These data suggest that WFS1, located at the nuclear periphery, has no/reduced access to enhancers of genes, when they are displaced from the nuclear envelope upon *Tmem38* KO (**new Supplementary Fig. 8d**). These data also convincingly demonstrate the specificity of the WFS1 ChIP.
 - 3) We now show new heatmaps demonstrating enrichment of WFS1 on active enhancers, compared to a flat signal on random genomic regions within open chromatin (**new Supplementary Fig. 9e**).
 - 4) We provide additional data, showing that the WFS1 ChIP protocol works properly, including Western blots demonstrating the presence of WFS1 in immunoprecipitated samples, as well as details on chromatin fragment size distributions (**new Supplementary Fig. 7a-d**).
 - 5) We show RPKM values, rather than log2 ratios in input and WFS1 ChIP samples, as these allow better visualization of WFS1 enrichment in specific genomic loci.
 - 6) We find high Pearson correlation coefficients when comparing ChIP-seq data sets of WFS1 in wildtype cells with that of ectopic WFS1-expressing WFS1 KO cells and show additional analyses of the ectopic WFS1 data set in the response below, confirming reproducibility of our ChIP analyses.
- We performed several additional qPCR and histone ChIP experiments to improve statistical analyses (**new Supplementary Fig. 3f, h**).
- We addressed all other points of the reviewers in the text and Figures.

POINT-BY-POINT RESPONSE TO REVIEWER COMMENTS

Reviewer #1 (Remarks to the Author):

In 'Peripheral localization of the MyoD1 gene is independent of heterochromatin tethers and mediated by binding of WFS1 to active answers', Gerogiou and co-authors use a novel reporter system in C2C12 cells to track the position of the MyoD1 gene locus as they manipulate both heterochromatin and potential anchoring proteins, ultimately implicating the gene WFS1 as contributing to anchoring the locus to the periphery via binding of distal MyoD1 enhancer elements. Their data on WFS1 binding throughout the genome suggests a potential role for it in binding enhancers more broadly and link it to expressed genes in myoblasts.

Given the significant role that nuclear and genomic organization plays in the process of myogenesis – not to mention the central role that MyoD1 plays in myogenesis - their finding is of significant interest, and will likely be of interest not only to those who work in myogenesis, but to those who consider re-organization of gene loci in other cell types as well. WFS1 has previously been described as playing a role in nuclear organization in myogenesis, but at the level of larger genomic regions, while the authors here delve into a single (critical) genomic locus.

The manuscript overall is logically laid out, with careful attention to controlled experiments. The materials and methods section is particularly thorough, and seems to provide all necessary information for reproduction of experiments. However, I do have a number of questions/suggestions for improvement as listed below – a few where I appear to have a different interpretation from the authors of some of the presented data that I'm hoping can be reconciled (points #4, 6, and 7), and the majority looking for more explanation on either rationale or context/interpretation. I have listed my points to reflect the order in which data and discussion were presented in the manuscript, rather than grouping by major/minor or a similar schema:

RESPONSE:

We thank the reviewer for the overall positive evaluation and the constructive comments, which we have all addressed.

1. The authors' approach of normalizing their radial distance analysis is predicated on correcting for maximal nuclear height, which the authors show to be essentially stable in their system in Suppl Fig 2. However, myocytes typically demonstrate elongation along at least one nuclear axis during differentiation (I believe that the authors would refer to this as the 'X' axis based on their model in Fig 1D). Can the authors comment (or show data on their decision process) on the choice to normalize to 'Z', as opposed to the axis that appears to show the greatest change during differentiation – 'X'? (I will say that, given the data shown in Suppl Fig 2f and 2g of their non-normalized data, I think the authors have likely chosen a reasonable approach, but I think some additional explanation of the logic would be helpful.)

RESPONSE:

To address the comment of the reviewer, we added more information on the analyses in the material and methods section (**page 38**), as explained in detail below:

We have chosen to normalize the measured closest distance of *MyoD1* locus to nuclear periphery based on the height, as the nuclei are always flat, meaning that the Z dimension (height) is always the smallest distance, compared to X and Y. We are therefore normalizing to the largest possible distance a locus can obtain from the nuclear periphery in 3D, which is always $z/2$. For instance, an internal position of a spot in the center of the nucleus would always have a maximum distance to the periphery of $z/2$ (see also Supplementary Fig. 1e). Even if nuclei would extend in X during differentiation, the maximum shortest distance to the nuclear periphery would remain $z/2$ as the elongation would cause a further decrease of the nuclear height.

We illustrate for the reviewer an example of a locus occupying an internal nuclear position displayed in orthogonal views below (**Figure 1 for reviewers**). The white dashed line represents the shortest distance to the nuclear periphery in each view. The XY view shows the shortest distance an internal spot would have if the analysis were performed in 2D, whereas the YZ and XZ views show the shortest distance to the periphery as detected in a 3D analysis.

Figure 1. Representative 3D reconstructed z stack of images showing intranuclear 3D position of a centrally located locus in proliferating myoblasts (green dots). White dashed lines indicate shortest distance in each two-dimensional view. Scale bar is indicated on image.

2. *Suppl Fig 1f-h: there is mention that the images in f – the light microscopy data – was obtained on a light microscope at 100x magnification but the it seems like numerous nuclei and cells are present per panel. Is it supposed to be the data in 1g (the confocal images) that was obtained at 100x?*

RESPONSE:

Supplementary Fig. 1f was generated with a 10x objective plus the 10x magnification of the ocular. We added this information in the legend to address the request of the reviewer.

3. *Figure 2 – is there a reason that Figure 2c (the Western blot) is referred to in text prior to the figure panels 2a and 2b? While I have no issue with the data in 2c, the flipped ordering made it somewhat confusing when reading.*

RESPONSE:

We rearranged the panels in Fig. 2 to make the western blot panel a.

4. *Figure 2 – does the comment in line 216 on page 10 about ‘applying a significance threshold of $p < 0.01$ ’ mean that when you apply a threshold of < 0.05 , you *do* see a difference between wildtype and Emerin KO in regards to MyoD1 locus positioning? Given that you also see a difference in the non-normalized distances plotted in Suppl Fig 3d, and that your other distance distributions show congruity between normalized and non-normalized distributions in both Figures 1 and 2, I would advocate that reporting a subtle increase in radial distance with Emerin KO would potentially be the more accurate way to describe this data (and I do not believe detracts significantly from the message of the manuscript).*

RESPONSE:

We agree with the reviewer and changed the significance threshold to $p < 0.05$ for all data in the manuscript. In the text describing Fig. 2b, we now state that we see a subtle increase in the radial distance of *MyoD1* to the nuclear periphery in Emerin KO cells. We added a * in the Figure and changed the text and the legend accordingly (**page 10**). Although we observed a slight increase in the distance of *MyoD1* from the nuclear periphery upon emerin depletion, we did not consider this subtle difference relevant given the high number of n and the relatively high p value compared to the p values obtained for WFS1 and Tmem38a knockouts (Fig. 4b). Furthermore, the LBR, Lamin A/C, Emerin triple knockout had no detectable effect on *MyoD1* nuclear localization (Fig. 2c).

5. *For the various figures that include the normalized radial distance distributions (eg. 1f and g, 2a and b), I think it would be helpful to the reader if, in addition to the ‘n’ of values analyzed, the mean values were reported on the figures (perhaps above each violin in the plot?) I found myself often considering where the mean line is located on the plots and comparing between conditions, and a numeric value would have been of help.*

RESPONSE:

We thank the reviewer for this suggestion to improve clarity. We have now added the median values on violin plots in all Figures that include normalized radial distances. However, we would like to state that we are not comparing medians but distributions (Kolmogorov-Smirnov test) in the statistical analyses as described in the methods (page 38).

6. For MyoD1 expression and the Histone ChIP shown in Suppl Fig 3 (panels e and g specifically), can the authors explain:

1) why there is so much more variability in the std deviation of expression of MyoD in wildtype cells compared to their knockouts? It seems that if that deviation was a bit lower in the wild-type samples that the authors would have seen statistically significant difference of expression (considering how small the standard deviation was between replicates in the KO cells)

RESPONSE:

Indeed, there was a bigger heterogeneity among the three biological replicates of the wildtype samples compared to the knockout samples. The variability originates from the difference in absolute expression levels of *MyoD1* between biological replicates. To improve statistical power, **we performed additional qPCR experiments** adding two more biological replicates. After adding these new data and performing batch normalization, we generated **new Supplementary Fig. 3h** and now state in the text that we do see a moderately significant difference between samples ($p=0.02$, **pages 11-12**).

and 2) Why there is nearly a 3-fold difference in mean signal for H3K4me3 in the MyoD1 promoter between wild type and KO cells? I have a hard time reconciling that difference with the statement in line 232-233 on page 10 that the histone ChIP signals were similar between KO and wild type cells.

RESPONSE:

We agree with the reviewer and added in the results section that we observe a reduction of the repressive H3K9me3 and the active H3K4me3 histone mark on the *MyoD1* promoter upon knockout of lamin A/C and LBR, in accordance with a slight reduction in *MyoD1* expression, likely due to the significant chromatin rearrangement upon lamin A/C and LBR depletion (**pages 11-12**).

We did not follow up on this observation in more detail, as anchorage of the gene was unchanged and our main focus was to identify mechanisms of *MyoD1* tethering. Generally, one would expect an upregulation of *MyoD1* expression combined with an increase in active histone marks upon loss of the heterochromatic environment at the nuclear periphery. Thus, the mechanisms seem to be more complex and probably indirect due to overall chromatin reorganization in knockout cells, which would require more in-depth analyses going beyond the scope of this study.

7. Related to the point above about H3K4me3 ChIP: the authors' statement in lines 253-254

on page 11 that their ChIP in LADs of H2K27me3 and H3K9me3 shows no difference between wild-type and KO cells seems surprising to me looking at the data in Suppl Fig 3g. Are those truly not different on statistical testing? The standard deviations look like they don't overlap, especially for H3K27. (I will note that I certainly agree with the authors' interpretation of the immunoblots showing no difference of overall histone mark levels in Suppl Fig 3f.)

RESPONSE:

The data shown in old Supplementary Fig. 3g are technical replicates. In order to address the concerns of the reviewer on the repressive marks, we performed **additional analyses** involving two biological replicates of repressive H3K9me2 and H3K9me3 marks on the LAD region and added this analyses as an **additional bar graph in new Supplementary Fig. 3g**. This analysis revealed no difference between wildtype and double knockout cells and indicates that the subtle increase observed in one of the biological replicates is likely due to technical variability (**pages 11-12**). In order to analyze epigenetic data in more detail we would have to perform more ChIP-seq experiments going beyond the scope of the study.

8. Figure 3g: Can the authors expand on why the reported n for MyoD locus analysis is so very variable between conditions? This figure panel stands out in particular to me : the n is 600 – 900 per condition here, and some other conditions have far lower numbers (for instance, nLAP1 KO in Suppl Fig 4h is only 19).

RESPONSE:

In the course of this study, we observed that changes in radial distance from the periphery can be detected even with a smaller number of data points. As no change in the position of *MyoD1* was observed upon LAP1 knockout in initial analyses with lower sample sizes, we chose not to generate additional data for a negative result and instead focused on other knockouts (see Supplementary Fig. 4h).

For analyses where we find changes upon knockout of genes, we repeated the experiments several times to obtain more biological replicates and increase the number of data points in order to convincingly support the main conclusions in the manuscript.

To support our explanation above, claiming that analyses with lower sample sizes yield similar distributions to those obtained with larger sample sizes, we show here for the reviewer a random downsampling of datasets shown in Fig. 3g and Fig. 4b (originally performed with large sample sizes $n > 500$) to $n = 20$ (**Figure 2 for reviewers**). While some differences are not statistically significant yet, clear differences in the distributions between wildtype and knockout cells can be observed, indicating localization changes of *MyoD1* even with smaller sample sizes.

Figure 2. (a, b) Normalized radial distance of a LAD region to the nuclear periphery in wildtype compared to lamin A/C and LBR double knockout cells as detected by 3D DNA-FISH, displayed as distributions of **(a)** complete dataset comprised $n_{wt}= 909$ and $n_{KO}= 666$ and **(b)** succeeding random downsampling comprised $n_{wt}= 20$ and $n_{KO}= 20$. **(c, d)** Normalized radial distance of *MyoD1* to the nuclear border in wildtype myoblasts versus cells with knockout of indicated proteins. Figure displays the distribution of *MyoD1* positioning in **(c)** complete dataset comprised $n_{wt}= 722$, $n_{4KO}= 454$ and **(d)** succeeding random downsampling comprised $n_{wt}= 20$ and $n_{4KO}= 20$. **** $p= 3.2 \times 10^{-10}$, *** $p_{4KO}= 2 \times 10^{-4}$. p-values of 2 downsampld datasets used for pairwise comparisons are indicated on the violin plots. H3K9me2-marked heterochromatin layer is shown as gray-shaded bar.

9. Figure 5e: As the authors mention in their text, I appreciate a shift between peak intensities when comparing the top and bottom panels (LAP2beta/WFS1 and Lamin A/C/WFS1, respectively), but as someone who does not typically consider this sort of data, the shift appears rather subtle to me. Can the authors give either some sort of statistical comparison or additional context to the relative strength of the difference between panels?

RESPONSE:

The small shift we see in these analyses is based on the fact that the distance between LAP2beta, an inner nuclear membrane protein, and lamin A/C, a member of the nuclear lamina underlying the inner nuclear membrane is very small (~5 nm) and below the resolution limit of high-resolution light microscopy. Based on these limitations, we did not attempt to get any quantitative analyses on distances but tested whether these two structures – nuclear membrane and underlying lamina - can be resolved in this setting. Thus, we can conclude that at this resolution we detect WFS1 at the inner nuclear membrane and slightly shifted compared to the intranuclear lamina structure. To make this clearer in the manuscript we added a larger magnification of the overlapping peaks as a **new inset in Fig. 5e** and modified the text in the results accordingly (**pages 15-16**).

10. Do the authors have a reference for their two-step crosslinking and increased sonication shearing approach or, alternatively, can they expand on the logic/evidence for that? I typically do not think about using that approach to ensure mapping signal to heterochromatic regions. And did the authors describe what size range their DNA was in after that extent of sonication? I didn't see comment on what size that resulted in.

RESPONSE:

The two-step cross-linking protocol has been found in several studies to improve the efficiency of chromatin immunoprecipitation of proteins not directly associated with DNA¹⁻³. We initially tried a single formaldehyde fixation step and failed to precipitate enough chromatin for sequencing. We would like to note that the two-step crosslinking process is known to require more sonication cycles for chromatin shearing compared to formaldehyde-only fixation⁴. We tested different sonication settings by performing fragment size assessment using agarose gel and Bioanalyzer analyses. Two sonication settings that displayed the desired distribution of fragment sizes on the agarose gel are shown below for the reviewer (**Figure 3a, b for reviewers**).

We have previously found that an increased number of sonication cycles favors immunoprecipitation of heterochromatic regions⁵. Therefore, to include both eu- and hetero-chromatic regions in our analysis and obtain chromatin fragment sizes typically used in ChIP-sequencing, we performed 2 subsequent sonication steps in the protocol, initially 30 cycles and, following the ChIP, an additional 6 cycles of sonication. An initial sonication of 36 cycles prior to the immunoprecipitation would lead to destruction of epitopes and hamper the efficiency of the pull-down⁴. A size-selection step is performed prior to NGS analysis selecting chromatin fragments ranging from 100 to 800 bps, which, following 30 sonication cycles, corresponds to 49% and 64% of the total sheared chromatin in the immunoprecipitated and input sample, respectively. To ensure that immunoprecipitated fragments above 800 bps are included in sequencing we added the additional sonication cycles and increased the content of fragments within the selected size range to 75% and 92%. To explain this protocol in more detail to the reader, we have included the fragment size distributions at the two sonication conditions in a **new Supplementary Fig. 7**. For all bioinformatic analyses we combined peaks obtained from samples sonicated for 30 and 36 cycles to include the different chromatin fractions.

We also describe this method in more detail including fragment size distributions in the results and methods (pages 16, 34, 36).

Figure 3. Fragment size assessment of sheared chromatin during optimization using (a) agarose gel and (b) the Bioanalyzer.

11. The authors report WFS1 peaks numbering in the ~10000 - 20000 range across the genome in Suppl Fig 7, and report a high association (69%) between WFS1 bound enhancers and expressed genes in Fig 7c. Can they comment on whether they are suggesting those 69% of expressed genes in myoblasts are located at the nuclear periphery, as implied by the WFS1 binding? And if so, does that comport with any available data (either from their own work or other sources) on where expressed genes in myoblasts are located in the nucleus?

RESPONSE:

We want to clarify here that we state that 68% of the genes linked to the **enhancers OVERLAPPING WFS1 peaks**, rather than ALL genes (these are only 12%), are expressed. Additionally, we propose that the enhancers rather than genes are located at the nuclear periphery.

In response to the reviewer, we checked several other genes, previously reported to localize at the nuclear periphery in C2C12 cells, namely *Myf5*⁶, *Dync1i1*, *Plxna2*, *Cxcl1* and *Vcam1*⁷. We found that these genes are expressed and included in the WFS1-associated enhancer-linked genes in C2C12 myoblasts. To demonstrate these new findings **we add new IGV browser tracks in Supplementary Figs. 8b, c**, showing the genomic loci of *Myf5* and *Plxna2* where WFS1 peaks coincide with several predicted enhancer elements. Furthermore, we performed **additional experiments** verifying the CHIP-sequencing data by CHIP-qPCR analysis of WFS1 enriched and non-enriched regions in *MyoD1* and *Plxna2* genes. New results are shown in **new Supplementary Fig. 8a, d** and in the results section (pages 17, 18, 19).

We also kindly refer the reviewer to our response to comment 11 of reviewer 2 below, in which we provide more examples of genes linked to WFS1-associated enhancers.

12. As mentioned by the authors, WFS1 has been previously implicated in mediating chromosomal position in myogenesis (Robson, et al. 2016) – can they specifically address whether that chromosome data helps shed light on their analysis of the MyoD1 locus (ie. is MyoD1 on one of the chromosomes analyzed specifically in that work?)

RESPONSE:

In Robson et al.⁷, the authors report that overexpression of WFS1 leads to increased peripheral localization of chromosome 5. Furthermore, they show increased peripheral localization of chromosomes 8, 10, 11 and 16 following the overexpression of another nuclear envelope protein, NET39. *MyoD1* is located on chromosome 7, therefore the data presented in Robson et al.⁷ do not provide insights for the positioning of the *MyoD1* locus. However, we want to point out that one cannot directly compare the analyses done by Robson et al., analyzing whole chromosomes, with our experiments focusing on a single genomic locus.

Reviewer #2 (Remarks to the Author):

This review is provided by Philippe Collas.

In this manuscript, Georgiou et al. report a novel mechanism of interaction of an expressed and differentiation-regulated gene locus to the nuclear envelope. This specifically involves, in C1C12 myoblasts, association of a distal (predicted) enhancer of the Myod1 gene with WFS1, an ER transmembrane protein which the authors also provide evidence of localization in the INM. Combinations of locus tagging, differential detergent extractions, imaging data (live, immunofluorescence and FISH) including a new pipeline for semi-automated quantification of locus positioning in the 3D nucleus, Ca9-mediated KOs and ChIPseq analyses collectively allow the authors to identify and characterize WFS1 as a novel regulator of locus (enhancer) tethering to the nuclear envelope.

This is a well conducted and informative study combining imaging and -omics experiments supported by many controls. The work significantly adds to our understanding of gene regulation mechanisms at the nuclear envelope, a notoriously largely repressive environment. The manuscript is essentially divided into two sections: 1) identification and characterization of the localization of WFS1, from imaging and associated approaches, and 2) identification of the genomic role WFS1 may have in myoblasts.

The authors should be commended on the carefulness of the imaging experiments, where several complementary methods are used (reporter cell line for live imaging, extractions/IFs, FISH, and quantifications). These experiments are loaded with relevant controls.

I however have two main sets of comments, which should be address to strengthen the authors' claims (points 1-7), and allow them to start testing their interesting hypothesis (discussion) on WFS1 being a regulator of Myod1 expression during differentiation, by perhaps modulating P-E interactions (points 8-9).

RESPONSE:

We thank the reviewer for the overall positive evaluation of our work and the constructive comments.

1. Fig 5, S6 and related text: p15, the authors should attempt to tie together their interpretations of their differential TX/digitonin extractions on the orientation and ER/INM localization of WFS1, with the high-res microscopy shown in Fig 5 and S6, from which they suggest an INM localization based on a shift in the WFS1 vs lamin A/C immunofluorescence signal. Their claim on the INM localization of WFS1 would be strengthened this way.

RESPONSE:

We are not sure what exactly the reviewer is suggesting with “tie together”. The digitonin- and Triton X- extractions clearly show that the N-terminus of WFS1 is in the cytoplasm/nucleoplasm, but it does not prove localization of WFS1 in the inner nuclear membrane, as the antibody signal in the digitonin-treated sample originates from the cytoplasmic localization of the N-terminus in the outer nuclear membrane and in the ER. High-resolution microscopy of in situ extracted samples is limited in resolution. In response to the reviewer, we summarized these data and conclusions based on it in the text (**page 16**).

2. Out of curiosity: why did the authors chose Pax7 as another locus to look at by imaging? What is known on Pax7 anchoring to the nuclear envelope?

RESPONSE:

We chose *Pax7* as a control as the locus is not associated with the nuclear periphery in proliferating myoblasts⁶. Indeed, we show in Fig. 1f and Supplementary Fig. 2f that we can clearly distinguish peripherally located *MyoD1* and *Pax7* in the nuclear interior in our assay. To address the comment of the reviewer we now add this information to the text (**page 7**). Furthermore, in subsequent studies that are beyond the scope of this study we plan to look at *Pax7* peripheral attachment in differentiated cells, as the *Pax7* locus was suggested to have the opposite localization pattern compared to *MyoD1*, localizing in the nuclear interior in proliferating myoblasts and at the nuclear periphery in differentiated cells.

3. My major concern is however on the WFS1 ChIP-seq data, Fig. 6 and associated supp Figs. Several points follow on that, which should be addressed. From the claims made on the genomic enrichment of WFS1, one might expect punctual enriched sites detectable merely with accumulated normalized read counts, as for TFs or sharp histone peaks. Out of curiosity, what do such profiles look like for WFS1?

RERESPONSE:

We thank the reviewer for bringing up this point. Following the suggestion of the reviewer, we now show normalized read counts for WFS1 and input, rather than log2 ratio WFS1/input in IGV browser images (WFS1 RPKM, Fig. 6a, **new Supplementary Figs. 8b, 8c**).

We feel that this allows a better detection of WFS1 enrichment on genomic regions. However, we also want to point out that we do not expect sharp peaks of enrichment as observed for transcription factors and histone marks, since the type of chromatin association of WFS1 may be very different from that of transcription factors. Similarly, other projects in our lab revealed genome-wide localization patterns of Swi/Snf nucleosomal remodeler complexes similar to that of WFS1.

In response to the reviewer's comment and to support our claim of a specific enrichment of WFS1 on active enhancers we now also provide new heatmaps of WFS1 signal on enhancers compared to the signal on random genomic regions located in constitutive inter-LADs (**new Supplementary Fig. 9e**).

4. WFS1 is an integral protein with 9 tm domains. Is WFS1 actually solubilized and detected in the chromatin fraction used for ChIP? Is it detected in the IP (done under ChIP conditions)?

RESPONSE:

In response to the reviewer's comment, we have added a WFS1 Western Blot of ChIP'ed wildtype and WFS1 knockout samples in **new Supplementary Fig. 7d**, clearly demonstrating that anti-WFS1 antibody brings down a specific band representing WFS1 homodimer⁸ under ChIP conditions, which is missing in the WFS1 KO sample.

5. Several points here: The WFS1 ratio profiles are noisy, and MACS2 calls peaks, it seems, not as much from WFS1 signal strength but mainly from (low) surrounding signal strengths; in other words, the profiles reveal many more signals, specific or not, than what is detected by MACS2 (Fig. 6a,b). Given the log₂ ratios shown, the ChIP-seq data themselves, and therefore peak calling, may not be extremely convincing.

RESPONSE:

Please see also comments on point 3 of this reviewer 2 related to noisy log₂ ratio profiles and the **newly added RPKM profiles** in IGV browser tracks.

As for MACS peaks, we used very stringent conditions when calling MACS peaks to minimize the risk of artifacts/unspecific peak calling. We initially called peaks using different softwares and different significance settings and visually investigated the resulting peaks by comparing them to the mapping signal and log₂ratio tracks of corresponding samples in the IGV browser. We finally chose the peak sets representing the signals best, with an FDR of 5% (minimum threshold of q-value to 0.05) and an average fold enrichment of peaks of 4.99 fold. 6,269 out of 18,283 peaks of the WFS1 30 sonication cycles sample, and 5,166 out of 12,674 peaks of the WFS1 36 sonication cycles samples showed enrichment values > 5-fold. Thus, our subsequent analyses were done with high confidence peaks.

Please also see our response to comment 1 of reviewer 3, in which we also provide several new analyses done with MACS peaks called with less stringent conditions, revealing similar overall results as shown in the manuscript, where MACS peaks were called with stringent conditions (see **Figure 8 for the reviewers** below).

So:

a. Is another WFS1 Ab available to validate the Ab used in the study?

RESPONSE:

Unfortunately, there are no other antibodies available that work in ChIP.

We show the reproducibility of the ChIP using this antibody, by comparing the ChIP of endogenous WFS1 with a ChIP of ectopic WFS1 expressed in WFS1 knockout cells and by determining the Pearson correlation coefficient. The Pearson correlation coefficient between the 30 cycle replicate pair was 0.81, and between the 36 cycle replicate pair 0.79.

We now report these data in the methods section (page 39).

Additionally, we have carefully validated the specificity of the WFS1 antibody by immunofluorescence and Western blot using WFS1 knockout cells as negative control (Fig. 5b and c).

b. The authors should validate their seq-data with by ChIP-qPCR in C2C12 using primers tiling “enriched” and non-enriched sites around the Myod1 gene and predicted enhancer sites (referring to Fig 6b). Can the seq profiles be replicated?

RESPONSE:

We thank the reviewer for this suggestion. We performed **new ChIP-qPCR experiments** demonstrating WFS1-enrichment on several active enhancer regions in *MyoD1* compared to regions depleted of WFS1 signal in ChIP-seq around the genes. We also directly compared two enhancer regions with high and low WFS1 log₂ratio signals found in ChIP-seq, corresponding to the positive and negative enhancer regions of *MyoD1*. Furthermore, we included similar WFS1-enriched and non-enriched regions around the *Plxna2* gene, reported in a previous study to be influenced by knockdown of WFS1⁷ (see **new Supplementary Fig. 8a**). These ChIP-qPCR analyses nicely confirm the specificity of the WFS1 ChIP-seq results showing significantly higher signal in enriched regions over the non-enriched ones as expected.

c. How much of the “noise” could be due to the double cross-linking? I assume the authors have tried only formaldehyde, and this was unsuccessful – is that so?

RESPONSE:

Yes, we also tried classical formaldehyde fixation and failed to ChIP chromatin efficiently using WFS1 antibody. Previous studies have shown that ChIP of proteins not directly associated with DNA require double crosslinking¹⁻³. Please see also our detailed response to comment 10 of reviewer 1.

6. It would be reassuring to see a ChIP-seq analysis of WFS1 in C2C12 cells KO for WFS1.

RESPONSE:

We agree with the reviewer and have performed WFS1 ChIP in several WFS1 knockout cell clones. However, when characterizing more closely the initial WFS1 knockout clones, in which we used sgRNAs targeting exon 5 and exon 8 (containing the transmembrane domains) in CRISPR/ Cas9-mediated knockout strategy, we noticed that some of the knockout cells still express some residual, probably small N-terminal fragment of WFS1 missing the transmembrane domains and locating in the cytoplasm and nuclear interior (**see Figure 4 a-d below for the reviewers**). In several attempts to obtain a complete WFS1 knockout cell clone we performed single cell cloning of the pooled knockout cells and further CRISPR/Cas9 mediated knockout using additional sgRNAs targeting WFS1 exon 2. However, TIDE (**Figure 4b for the reviewers**, left panel) and qPCR analysis revealed that around 25% of these cells carry an in-frame deletion and still express low levels of a small, potentially N-terminal, WFS1 fragment that is presumably detected by the WFS1 antibody. Accordingly, WFS1 ChIP-qPCR in these WFS1 KO cells brings down some chromatin (**Figure 4e for reviewer**), but IMPORTANTLY, we did no longer see specific enrichment on active versus inactive enhancer regions around *MyoD1* and *Plxna2* as observed in wildtype cells (compare to Supplementary Figure 8a in the manuscript). Thus, we concluded that a remaining lowly expressed small N-terminal fragment in WFS1 KO cells, which localizes also to the nuclear interior, is precipitated by the WFS1 antibody and brings down chromatin non-specifically. Based on these findings, performing ChIP-seq in WFS1 knockout cells is not feasible.

However, we want to point out that we performed **several additional experiments** and added **several new Figures** in the manuscript, which all clearly and convincingly demonstrate specificity of the WFS1 ChIP. These include WFS1 ChIP-qPCR experiments on *MyoD1* loci and other peripheral gene loci, confirming binding to active but not to inactive enhancers (see our response to this reviewer's point 5b above and our **new Supplementary Fig. 8a, b, c**).

Furthermore, we have performed a WFS1 ChIP in *Tmem38* knockout cells (see our response to point 8 of this reviewer below), convincingly showing specificity of the WFS1 ChIP.

As another support for specificity of the WFS1 ChIP in WT cells, we demonstrate that the WFS1 ChIP signal is enriched on WFS1 peak-overlapping enhancers compared to overlapping promoters and non-overlapping enhancers (see Supplementary Fig. 9d). Similarly, in Fig. 7b we show a significant overlap of WFS1 peaks with candidate cis-regulatory elements from ENCODE, while randomized peaks barely overlapped. Furthermore, the WFS1 ChIP signal is clearly detected on WFS1 peak-overlapping enhancers, while no enrichment is found on random genomic regions within constitutive inter-LADs (**new supplementary Fig. 9e**).

Altogether, these results demonstrate that the WFS1 ChIP is specific and reliable, revealing WFS1 enrichment on a subset of active enhancers.

Figure 4. (a) Schematic representation of the mouse *WFS1* genetic locus including 8 exons (brown boxes) and introns (angled lines). Regions targeted by synthetic sgRNAs in exons 2, 5 and 8 are indicated by orange stars. Light brown boxes designate predicted transmembrane domains. Arrows indicate primer pairs used for RT-qPCR. **(b)** Targeting efficiency was determined using the TIDE software. The region spanning the editing sites in exon 2 (left) and exon 8 (right) was amplified from genomic DNA of *WFS1* knockout and wildtype cells by PCR. PCR products were sequenced and uploaded in TIDE. Position 0 shows percentage of unedited wildtype sequences, R2 value demonstrates the goodness-of-fit, p values are calculated by two-tailed t-test of the variance-covariance matrix of the standard errors. Yellow arrow indicates the presence of an in-frame deletion variant (-21 bps). **(c)** *WFS1* transcript levels in knockout relative to wildtype cells were analyzed by RT-qPCR at different positions of the transcript (using primer pairs indicated in panel a). Data represent mean values \pm SEM from 3 technical replicates. **(d)** *WFS1* protein levels were assessed by immunofluorescence analysis in wildtype and knockout cells. Scale bars, 10 μ m. **(e)** *WFS1* ChIP-qPCR analysis in *WFS1* knockout myoblasts of regions enriched (*MyoD1* CE, *Plxna2* enhancer, positive control) or depleted (*MyoD1* DRR, *Plxna2* downstream, negative control) for *WFS1* signal in *WFS1* ChIP-seq analysis of wildtype cells. Bar graph displays mean \pm SEM from 2 biological replicates. Horizontal dashed line indicates signal obtained using unspecific IgG antibody in control ChIP.

7. What is the chromosome by chromosome distribution of WFS1 ChIP-seq signals and peaks? Is WFS1 enriched (given the type of analysis done) on specific (more peripheral?) chromosomes? Related to this: if WFS1 is an INM protein, how would the authors explain a distribution on (all?) chromosomes?

RESPONSE:

We have checked the distribution of WFS1 peaks over individual chromosomes and found roughly equal distribution over all chromosomes, which is very similar to the distribution of cLADs over all chromosomes (**see Figure 5 for the reviewer**). Thus, both peripherally localized WFS1 bound genomic regions, as well as peripherally localized cLADs are distributed over all chromosomes.

We would like to point out that, in our opinion, nuclear positioning of whole chromosome cannot be compared directly with that of specific genomic regions. If a chromosome is found mostly in the nuclear interior, small genomic regions on this chromosome can still extend towards the periphery. We also added a brief statement that WFS1 peaks are distributed over all chromosomes in the text (**page 19**).

Figure 5: Bar graph shows distribution of WFS1 peaks and constitutive LAD regions (cLADs) over individual mouse chromosomes displayed as percentage of the total number identified genome wide.

8. Ideally, it would be good to see WFS1 ChIP-seq data under a condition where WFS1 is delocalized from the INM; if I am not mistaken however, this may be tricky since even a KO of *Tmem38a* does not delocalize WFS1 from the INM. However: since KO of *Tmem38a* delocalizes *Myod1* from the nuclear periphery (their imaging data), it would be good to test by ChIP-qPCR of WFS1 if, in *Tmem38a* KO cells, the *Myod1* gene and enhancer(s) are dissociated from WFS1. This would strengthen the role of WFS1 in anchoring *Myod1*.

RESPONSE:

We thank the reviewer for suggesting this interesting experiment in order to confirm specificity of the WFS1 ChIP. According to his suggestion, we performed WFS1 ChIP (ChIP-qPCR analyses) in Tmem38 knockout cells, in which *MyoD1* is displaced from the nuclear periphery.

Reassuringly, we observe a significant reduction in the binding of peripheral WFS1 to the *MyoD1* core enhancer region, as well as to the enhancer region of another WFS1-associated gene, *Plxna2*, in the Tmem38 knockout versus wildtype cells. We now show these important new results in **new Supplementary Fig. 8d** and in results section (**pages 18, 19**).

9. To provide relevant additional epigenetics background around Myod1, it would be informative to establish the H3K9me3 profile of C2C12 cells, from public or own ChIP-seq data. In particular, the heterochromatin landscape in the Myod1 region upstream of Myod1 and therefore of the LAD reported in Fig 6a would be useful to see. H3K9me3 should be included in the analyses (Fig 7 and relevant supp Figs).

RESPONSE:

We added the H3K9me3 signal in the IGV browser views shown in **Fig. 6a** and in **new Supplementary Figs. 8b, c**, and added heat maps of the H3K9me3 signal on WFS1 ChIP peaks in **new Supplementary Fig. 9c**.

As we did not find any publicly available H3K9me3 dataset with reasonable number of peaks called (the ones available on CISTROME showed very low number of peaks) we were not able to analyze the signal of WFS1 ChIP-seq on H3K9me3 peaks in heatmaps directly.

However, during revision and evaluation of the cLAD regions it came to our attention that the cLAD annotation used to generate the IGV browser images was outdated due to an update of the mouse genome assembly (from mm9 to mm10), and was different from the correct cLAD annotation (with coordinates converted from mm9 to mm10) used for all other analyses in the manuscript. Accordingly, we have now replaced the cLAD tracks in the IGV browser images (**Fig. 6A, new supplementary Fig. 8b, c**) with the correct annotation and in doing so, the former cLAD right next to the *MyoD1* locus was not displayed anymore in the new annotation. ChIP-qPCR analysis of that region compared to a confirmed cLAD verifies that the epigenetic landscape matches that of a more “active” region (**see Figure 6 for reviewer**).

Figure 6. ChIP-qPCR analyses of indicated active (H3K27ac, H3K4me3) and repressive (H3K27me3, H3K9me3) histone marks in the region downstream of *MyoD1* (*MyoD1* downstream) and a LAD region (LAD).

10. They authors raise the interesting hypothesis of WFS1 being a regulator of Myod1 expression and P-E interactions. Fully testing this idea would require doing HiC or the like in myoblasts and myotubes, which may be beyond the scope of this study. However, the idea should be addressed by a ChIP-seq analysis of WFS1 and relevant histone marks (H3K27ac at least; H3K9me3) in myotubes. Should HiC or other enhancer prediction data also be available for differentiated C2C12 cells (myotubes), a similar analysis of what the authors report in myoblasts should be carried out. Altogether, these experiment would enhance the significance of their findings.

RESPONSE:

As the reviewer mentions here, as well as in the first paragraph of his review, going forward with analyzing WFS1 ChIP in differentiated myotubes and comparing it to HiC data is a logical follow-up study, but clearly goes beyond the scope of this study, which focuses on peripheral tethering mechanisms of WFS1 in proliferating cells.

In order to provide a first preliminary analysis of myotubes for the reviewer, we show in **Figure 7 for the reviewer below** the epigenetic landscape of *MyoD1* and its super-enhancer region in myotubes versus myoblasts. We see an increased occupancy of active H3K4me3 and H3K27ac histone marks in myotubes compared to myoblasts on *MyoD1* enhancer regions, which is expected given the upregulation of MyoD1 expression during differentiation.

However, we found no indication of novel/different regions carrying enhancer-specific marks and have no reason to believe additional enhancer regions are present in differentiated cells in this genomic region. Hence, we did not create myotube-specific enhancer prediction maps, as this would need more in-depth analyses going beyond the scope of this study.

Figure 7. IGV tracks of the mouse genomic region (mm10) containing *MyoD1* and its regulatory region (35 kb upstream). Included are ChIP-seq signal tracks of histone modification H3K4me3, H3K27ac, H3K4me1 and H3K27me3 in myoblasts (MB) and myotubes (MT) as well as gene annotations from NCBI reference sequence database. Known cis-regulatory regions of *MyoD1* are depicted at the bottom (drawing not to scale). CE: core enhancer, DRR: distal regulatory region, PRR: proximal regulatory region.

11. The authors extend their findings beyond *MyoD1* by providing GSEA terms of expressed genes whose predicted enhancers are also, it seems, bound by WFS1. Providing 1-2 more examples, even only through a browser views, would be interesting.

RESPONSE:

We thank the reviewer for this interesting suggestion. In addition to *MyoD1*, we now tested another gene shown by Robson et al.⁷ to localize at the periphery in myoblasts (*Plxna2*), as well as *Myf5*, a known regulator of muscle cell differentiation **by additional ChIP-qPCR experiments**. Importantly, we observe enrichment of WFS1 ChIP signal at the predicted enhancer regions for these genes. We provide IGV browser views of these genomic loci in **new Supplementary Figs. 8b, c)** and describe the finding in the text (**page 18**). Additionally, we add IGV browser images of a couple of additional potential peripheral genes⁷ (*Cxcl1* and *Vcam1*) in **Figure 8 for the reviewer** below.

Figure 8. (a, b) IGV tracks of mouse genomic regions (mm10) containing genes suggested to associate with the nuclear periphery in C2C12 cells and their predicted regulatory elements. **(a)** *Cxcl1* and **(b)** *Vcam1* locus. Included are WFS1 ChIP tracks (RPKM, reads per kilobase per million) following sonication for 30 cycles, WFS1 peak regions identified by MACS, constitutive LAD regions (cLADs), ChIP-seq signal tracks of histone modification H3K4me3, H3K27ac, H3K4me1, H3K27me3 and H3K9me3, RNA polymerase II (Pol II) ChIP-seq signal track, enhancer predictions and gene annotations from NCBI reference sequence database gene track.

12. Could the authors speculate on whether WFS1 would also be involved in such tethering functions in other cell types / lineages, based on known expression profiles?

RESPONSE:

We now add a paragraph in the discussion addressing this comment (**page 24**).

WFS1 is known to be abundantly expressed in other cell types/tissues, such as brain and pancreatic islets⁹, with a predominant localization in the ER and a proposed function in intracellular Ca²⁺ homeostasis regulation¹⁰ and ER stress response⁸. While a tethering function in other cell types cannot be excluded with certainty, WFS1 has only been proposed to localize to the nuclear membrane of muscle cells so far⁷.

13. Lastly, out of curiosity, could the authors try to speculatively unify the role(s) of WFS1 in the INM, both as a regulator of Ca homeostasis and as enhancer tether? It is interesting that several proteins of the INM, e.g. LBR, have enzymatic and anchoring functions.

RESPONSE:

We add a paragraph speculating on the dual role of WFS1 as anchor for enhancers and as a calcium regulator in the discussion (**page 24**).

It is interesting to note here that since Ca²⁺ signaling is involved in differentiation¹¹ it is tempting to speculate that Ca²⁺-mediated changes at the nuclear envelope could potentially regulate, at least in part, enhancer and/or gene tethering.

Reviewer #3 (Remarks to the Author):

Georgiou and colleagues investigate the role of mechanisms underlying positioning of the MyoD1 locus in myoblasts. They use a combination of DNA FISH, immunofluorescence, and occupancy studies in a MyoD1 reporter line to describe the role of a nuclear envelope protein, WFS1, in mediating MyoD1 positioning. The authors suggest that WFS1, an ER membrane protein, and not heterochromatin-related mechanisms, is responsible for peripheral tethering of MyoD1. Overall, the manuscript is well-written and data is presented well. The mechanisms by which loci get tethered get to different nuclear landmarks (in this case the nuclear lamina) is of interest. However, several conceptual and technical flaws dampen my enthusiasm substantially (outlined below) for the current version.

RESPONSE:

We thank the reviewer for the overall positive evaluation of our work and the constructive comments.

1. The WFS1 occupancy data is very difficult to interpret. First, the signal/tracks shown in Fig 6a appear to be very noisy. The signal appears widespread and the logic behind MACS peaks identifying areas of local signal enrichment is not clear. Moreover, how the replicates compared to each other in terms of peaks and signal intensity across the genome, should be presented. Given no occupancy data has been previously been published for WFS1 – the data would be much more convincing if a negative control was included (WFS1 ChIP from WFS1 KO cells or WFS1 ChIP in conditions in which a competitive peptide to the ab is introduced).

RESPONSE:

a) As for the noisy signal, please see also response to reviewer 2, points 3 and 5.

In Fig. 6a upper panel we show large genomic regions, where the log₂ ratio is illustrated in fixed bins, making it difficult to clearly detect WFS1-enriched regions. In a zoomed-in region (Fig. 6a lower panel), WFS1-enriched and signal-free genomic regions can be detected much more clearly. We want to point out that we do not expect sharp peaks of enrichment as

observed for transcription factors and histone marks, since the type of chromatin association of WFS1 may be very different from that of transcription factors. To address the concerns and following the suggestion of reviewer 2, we now show normalized read counts for WFS1 ChIP and input in IGV browser image (RPKM) instead of log₂ ratios (ChIP/input) in **new, adjusted Fig. 6a**. We feel that this view makes it easier to appreciate WFS1-enriched versus non enriched regions.

b) As for MACS peaks, please also see our response to reviewer 2, point 5. We used very stringent conditions when calling MACS peaks to minimize the risk of artifacts. Thus, our subsequent analyses are done with high confidence peaks. We show **in Figure 9 for the reviewers** below MACS peaks called with less stringent settings (q value=0.3 versus q value=0.05 in original dataset) revealing more peaks (roughly 2.5 fold increase in peak number) and find that the main conclusions are not changing, if we use these peaks for further analyses (e.g. epigenetic mark signal on peaks, overlap with distal enhancer elements, compare to Fig. 7c and Supplementary Fig. 9c). Thus, doing the analyses at very stringent conditions, as done in the manuscript, avoids potential artifacts, but importantly, does not hide any additional information, which would potentially be revealed in analyses done under less stringent conditions.

Figure 9. (a) IGV tracks of mouse genomic regions (mm10) containing the *MyoD1*, *Myf5* and part of the *Plxna2* loci. Included are WFS1 ChIP tracks (RPKM, reads per kilobase per million) following sonication for 30 cycles, WFS1 peak regions identified by MACS using a q value of 0.05 and 0.3, constitutive LAD regions (cLADs) and gene annotations from NCBI reference sequence database gene track. **(b)** Average binding profiles and heat maps displaying enrichment of ATAC-seq, H3K27ac,

H3K4me1, H3K4me3, H3K27me3 and H3K9me3 CHIP-seq signal on WFS1 MACS peaks (q value=0.3) \pm 3 kb from center. Heat maps are ranked in descending order of CHIP-seq signal intensity. **(c)** Degree of overlap (%) of WFS1 MACS peaks (q value=0.3) with candidate cis-regulatory elements from ENCODE (left). Overlap with randomized peaks is shown as control (right). dELS: distal Enhancer-Like Sequences, pELS: proximal Enhancer-Like Sequences, PLS: Promoter-Like Sequences.

c) As for the replicates, we now show evidence for the reproducibility of the CHIP by determining the Pearson correlation coefficient of a CHIP of endogenous WFS1 and a CHIP of ectopic WFS1 expressed in WFS1 knockout cells. The Pearson correlation coefficient between the 30 cycle replicate pair was 0.81, and between the 36 cycle replicate pair 0.79. Data were initially only provided in the summary report form and **we now add them to the methods section** of the manuscript as well **(page 39)**.

In order to support our conclusion, we show here for the reviewer an IGV browser view of the WFS1 CHIP-seq RPKM signal as detected around the *Ptxna2* gene region in the two replicates (**see Figure 10 for the reviewer** below). We additionally show that the ectopic WFS1 CHIP signal is enriched in open regions carrying active epigenetic modifications and the epigenetic landscape on the ectopic WFS1 CHIP peaks is abundant in active marks and depleted of repressive marks. Thus, the analyses of the WFS1 data set in the 2nd replicate yielded highly similar results compared to the analyses of the 1st replicate shown in the manuscript (Fig. 7a and Supplementary Fig. 9c).

Figure 10. (a) IGV tracks of mouse genomic region (mm10) containing part of the *Plxna2* locus. Included are WFS1 ChIP tracks (RPKM, reads per kilobase per million) of wildtype and ectopic WFS1 following sonication for 30 and 36 cycles, WFS1 peak regions identified by MACS using a q value of 0.05, as well as gene annotations from NCBI reference sequence database gene track. **(b)** Average binding profiles and heat maps showing enrichment of ectopically expressed WFS1 in WFS1 knockout background (\log_2 [ChIP/input]) on ATAC-seq, H3K27ac, H3K4me1 and H3K4me3 ChIP-seq peaks \pm 3 kb from center. Heat maps are ranked according to WFS1 enrichment in descending order. **(c)** Average binding profiles and heat maps displaying enrichment of ATAC-seq, H3K27ac, H3K4me1, H3K4me3, H3K27me3 and H3K9me3 ChIP-seq signal on ectopic WFS1 MACS peaks \pm 3 kb from center. Heat maps are ranked in descending order of ChIP-seq signal intensity.

d) as for the negative control, please see also our response to reviewer 2, point 6 and **Figure 4 for the reviewers above**).

We tried hard to generate single WFS1 knockout cell clones to perform WFS1 control ChIP. However, even when targeting the WFS1 locus with a combination of 4 different sg RNAs in the CRISPR/Cas9 knockout strategy, targeting exons 2, 5 and 8, we still detected the presence of a low abundant, small, potentially N-terminal WFS1 fragment in these knockout clones, which localizes to the cytoplasm and the nuclear interior. WFS1 ChIP analyses in these knockout clones brought down some chromatin, but importantly, we did not see any more the enrichment of signal on active versus inactive enhancer regions of *MyoD1* and another peripheral gene, *Plxna2*, as seen in wild type cells (**compare Figure 4 for reviewer above with new Supplementary Figs. 8a in manuscript**). Thus, performing ChIP-seq of the WFS1 knockout cells seems unfeasible.

However, we performed several additional experiments and added several new data to show specificity of the WFS1 ChIP-seq.

Most importantly, we performed WFS1 ChIP (ChIP-qPCR analyses) in *Tmem38a* knockout cells, in which *MyoD1* is displaced from the nuclear periphery. Reassuringly, we observe a significant reduction in the binding of peripheral WFS1 to the *MyoD1* core enhancer region, as well as to the enhancer region of another WFS1-associated gene, *Plxna2*, in the *Tmem38a* knockout versus wildtype cells. We now show these important new results in **new Supplementary Fig. 8d** and in results section (**pages 18, 19**).

Additionally, we performed **new ChIP-qPCR experiments** revealing WFS1-enriched regions on *MyoD1*- and *Plxna2*-linked enhancers compared to non-enriched regions around the genes and show additional IGV browser images revealing enrichment of WFS1 on the active enhancers of *Plxna2* and *Myf5*, confirming the results of the ChIP-seq analyses (see **new Supplementary Fig. 8a, b, c**).

Altogether, these additional analyses provide strong evidence for the specificity of the WFS1 ChIP analyses.

2. Related to the ChIP-seq data, while the MACs peaks show a potential enrichment in enhancers – what fraction of signal is found in enhancers? Based on the tracks, only a small amount would be, which makes claims regarding enhancers being enriched for WFS1 occupancy tenuous.

RESPONSE:

In Supplementary Fig. 9d we compare the WFS1 signal on WFS1 peaks-overlapping enhancers (10.05% of all enhancers) versus non-overlapping enhancers as well as on overlapping promoters versus overlapping promoters. Given that the high confidence WFS1 MACs peaks are covering only 0.38% of the genome (see Supplementary Fig. 7e), it is not feasible to calculate the fraction of the total WFS1 signal on enhancers (comparison of total signal on enhancers versus total signal on genomic region outside enhancers). Identification of genomic regions significantly enriched for specific proteins is therefore usually done by peak calling algorithms. Notably, using less stringent settings for peak calling, resulting in increased number of peaks and genome coverage, yields similar results leading to similar

main conclusions, observing an enrichment of WFS1 signal in active enhancer regions (please see response to comment 1c of this reviewer).

3. I would encourage the authors to show more of their FISH quantification data as absolute distance from the nuclear periphery/lamina as opposed to the percentage measure they use most often. The absolute distance provides better context for how far the locus is from the periphery as opposed to the current percent data. In addition, how the authors measured H3K9me2 thickness is not clear.

RESPONSE:

In response to the reviewer's suggestion, we now show absolute rather than normalized data of the FISH experiments in **new Fig. 3g**.

As for the determination of H3K9me2 thickness, we explain the analysis in more detail in a **new chapter in the methods section (page 31)**. Additionally, we add points indicating the regions used for measuring the thickness of the H3K9me2 signal in adjusted **Fig. 1g**.

4. Based on the authors' current work, it would be helpful if they could nominate other loci that may be WFS1 responsive in terms of positioning? This would strengthen the impact of the paper, as currently it is unclear if MyoD1 is the only locus whose positioning is dependent on WFS1.

RESPONSE:

We now show IGV browser views of regions around other genes shown to be influenced by WFS1 knockdown in Robson et al⁷ (*Plxna*, *Myf5*) (**new Supplementary Fig. 8b, c**), as well as two additional loci (*Cxcl1* and *Vcam1*) in **Figure 8 for reviewers** here in our response to comment 11 of reviewer 2. Additionally, we performed **new** ChIP-qPCR experiments showing binding of WFS1 to an active enhancer in the *Plxna2* locus (**new Supplementary Fig. 8a**). Overall, these data show that WFS1-mediated tethering to the nuclear periphery is a more general mechanism not restricted to *MyoD1*.

5. The WFS1 localization data is primarily based on exogenous expression. While it is well-done, it is still difficult to interpret because wouldn't most proteins which are over-expressed show some accumulation/trafficking in the ER? Have the authors considered endogenous ab staining and/or knocking a GFP tag into the WFS1 locus?

RESPONSE:

We also show specific staining of the endogenous WFS1 in Fig. 5b, including a negative control (WFS1 KO cells). These analyses show that the WFS1 antibody is specific and that the endogenous protein is also localized mainly to the ER. Therefore, a diffusion of overexpressed protein into the ER is not a problem in this setting.

We also tried high-resolution imaging using endogenously stained protein but this did not produce clear signals due to the overall low staining intensity and therefore increased the risk of artefacts. Knocking in GFP is a very good suggestion and will be done in follow up experiments, but is beyond the scope of this study.

6. In the LMNA and LBR KO studies (Fig 3), the authors conclude that LADs move away from the periphery based on 3G. However, the probe is infrequently at the periphery in the WT cells based on the plots provided? Hence, perhaps it is expected that the positioning is not LMNA/LBR-dependent.

RESPONSE:

Our data in Fig. 3g show a clear difference of LAD positioning in Lamin A/C and LBR double knockout versus wildtype cells. We do not expect that all LADs are detected to be at the periphery as it has been shown that only 30% of all LADs are localized at the periphery in a given cell and that many LADs are reshuffled to and from the periphery during the cell cycle¹². However, analyzing a large number of individual cells (n > 600) in FISH experiments is sufficient to reveal a clear statistically significant difference in knockout versus wildtype cells.

References

- 1 Texari, L. *et al.* An optimized protocol for rapid, sensitive and robust on-bead ChIP-seq from primary cells. *STAR Protoc* **2**, 100358 (2021). <https://doi.org:10.1016/j.xpro.2021.100358>
- 2 Nowak, D. E., Tian, B. & Brasier, A. R. Two-step cross-linking method for identification of NF-kappaB gene network by chromatin immunoprecipitation. *Biotechniques* **39**, 715-725 (2005). <https://doi.org:10.2144/000112014>
- 3 Zeng, P. Y., Vakoc, C. R., Chen, Z. C., Blobel, G. A. & Berger, S. L. In vivo dual cross-linking for identification of indirect DNA-associated proteins by chromatin immunoprecipitation. *Biotechniques* **41**, 694, 696, 698 (2006). <https://doi.org:10.2144/000112297>
- 4 Carter, B., Dykhuizen, E., Gerrin, S. & Khoja, H. Optimizing a Dual Fixation Protocol to Study Protein Complexes Binding to Chromatin in vivo.
- 5 Gesson, K. *et al.* A-type lamins bind both hetero- and euchromatin, the latter being regulated by lamina-associated polypeptide 2 alpha. *Genome Res* **26**, 462-473 (2016). <https://doi.org:10.1101/gr.196220.115>
- 6 Demmerle, J., Koch, A. J. & Holaska, J. M. Emerin and histone deacetylase 3 (HDAC3) cooperatively regulate expression and nuclear positions of MyoD, Myf5, and Pax7 genes during myogenesis. *Chromosome Res* **21**, 765-779 (2013). <https://doi.org:10.1007/s10577-013-9381-9>
- 7 Robson, M. I. *et al.* Tissue-Specific Gene Repositioning by Muscle Nuclear Membrane Proteins Enhances Repression of Critical Developmental Genes during Myogenesis. *Mol Cell* **62**, 834-847 (2016). <https://doi.org:10.1016/j.molcel.2016.04.035>
- 8 Wang, L. *et al.* WFS1 functions in ER export of vesicular cargo proteins in pancreatic beta-cells. *Nat Commun* **12**, 6996 (2021). <https://doi.org:10.1038/s41467-021-27344-y>
- 9 Takeda, K. *et al.* WFS1 (Wolfram syndrome 1) gene product: predominant subcellular localization to endoplasmic reticulum in cultured cells and neuronal expression in rat brain. *Hum Mol Genet* **10**, 477-484 (2001). <https://doi.org:10.1093/hmg/10.5.477>

- 10 Nguyen, L. D. *et al.* Calpain inhibitor and ibudilast rescue beta cell functions in a cellular model of Wolfram syndrome. *Proc Natl Acad Sci U S A* **117**, 17389-17398 (2020). <https://doi.org:10.1073/pnas.2007136117>
- 11 Sinha, S., Elbaz-Alon, Y. & Avinoam, O. Ca²⁺ as a coordinator of skeletal muscle differentiation, fusion and contraction. *Febs j* **289**, 6531-6542 (2022). <https://doi.org:10.1111/febs.16552>
- 12 Kind, J. *et al.* Single-Cell Dynamics of Genome-Nuclear Lamina Interactions. *Cell* **153**, 178-192 (2013). <https://doi.org:https://doi.org/10.1016/j.cell.2013.02.028>

SUMMARY OF REVISION 2

We are pleased to see that two reviewers are fine with our revisions and do not request additional experiments. In the revision 2 we now address the remaining concerns of reviewer 3 on the quality of the ChIP.

We performed new WFS1 ChIP experiments in single cell clones of WFS1 and Tmem38a knockout cells and, additionally, we performed all requested bioinformatic analyses of the ChIP replicates as suggested by the reviewer. Together, these experiments and analyses show specificity and reproducibility of our ChIP data. We added the following **new Supplementary Figures and data** to the manuscript:

- New Supplementary Fig. 8b, revealing WFS1 enrichment on active *MyoD1* enhancer in WFS1 wildtype, but not in WFS1 knockout single cell clones.
- New Supplementary Fig. 8c, revealing loss of WFS1 enrichment on active *MyoD1* enhancer in Tmem38a knockout cells, in line with the release of the *MyoD1* locus from the nuclear periphery upon Tmem38a knockout.
- New Supplementary Fig. 7e, showing a summary of called peaks from WFS1 ChIP replicates at the two conditions used (30 and 36 sonication cycles).
- New Supplementary Fig. 7f, showing the specific WFS1 ChIP signal on the called peaks, while respective input controls had only background signal on the same regions.
- New Supplementary Fig. 7g, showing similar WFS1 enrichment on the called WFS1 peaks in the replicates.
- New Supplementary Fig. 9e, showing specific enrichment of WFS1 on active enhancers overlapping WFS1 peaks, compared to a flat background signal on non-overlapping enhancers.
- Updated versions of Supplementary Figures 3 and 5 to address minor points of the reviewer.

We updated the Materials and Methods section and source data to accommodate new data sets shown in the new Figures.

Please find our detailed responses to the comments of reviewer 3 below.

We hope that with these additional data and new comprehensive analyses we can convince the reviewer about the quality and specificity of our ChIP. In order to allow the reviewer to check all ChIP data shown in the manuscript, including ChIP signals of the replicates as well as called peaks, we **uploaded the data on the UCSC server** and provide the link to access the data in the point-by-point response file.

POINT-BY-POINT RESPONSE TO REVIEWERS' COMMENTS

Reviewer #2 (Remarks to the Author):

Georgiou et al. now provide a substantially revised and improved manuscript. All my comments have been addressed thoroughly with an impressive array of additional experiments, additional controls at various levels, and refined conclusions and interpretations. I have no further comment and find the manuscript acceptable for publication.

RESPONSE:

We are pleased that reviewer 2 is satisfied by our response to her/his comments in revision 1.

Reviewer #3 (Remarks to the Author):

This is a revised manuscript by Georgiou and colleagues describing the role for WFS1 in mediating positioning of Myod1 relative to the nuclear periphery (though it is unclear if WFS1 KD affects myogenesis). Reviewer 2 and myself both raised concerns about the WFS1-ChIP-seq datasets. While the ChIP-qPCR is supportive and I appreciate the additional technical data provided, I remain concerned about the quality, accuracy and interpretability of the WFS1 ChIP-seq. Suggestions to strengthen the analysis of the datasets:

RESPONSE:

To address the concerns of the reviewer, we performed new WFS1 ChIP-qPCR experiments in a WFS1 knockout single cell clone, revealing only background signal in WFS1 knockout compared to wildtype cells. In addition, we performed several additional bioinformatic analyses of ChIP replicate data sets. These new data (see below) support quality and interpretability of the ChIP-seq data.

1. While CRISPR did not work to KO the gene, the authors could consider siRNA to reduce levels and then perform ChIP-seq. While similar peaks were called at different thresholds (and this is supportive that the peak calling method is ok), it does not address the signal:noise characteristics of the chip-signal.

RESPONSE:

As siRNA-mediated knockdown of WFS1 did not work efficiently, we isolated single WFS1 knockout cell clones from the mixed cell population previously used. WFS1 ChIP-qPCR analyses in the single cell clone, including 3 biological replicates, revealed significant reduction of WFS1 binding to the active *MyoD1* enhancer in WFS1 knockout versus wildtype cells (**see new Supplementary Fig. 8b**). In addition, we generated additional biological replicates of the WFS1 ChIP in the *Tmem38a* knockout cells, in which *MyoD1* is detached from the nuclear periphery (**new Supplementary Fig. 8c**). As expected, WFS1 enrichment was reduced on the *MyoD1* enhancer in *Tmem38a* knockout versus wildtype cells. Together with the WFS1 ChIP-qPCR analyses in wildtype cells, revealing clear enrichment of WFS1 on active enhancers versus only background signal outside of these enhancers (see Supplementary Fig 8a), these analyses show specificity of the WFS1 ChIP signal on active enhancers.

Methods sections and source data were adjusted to accommodate new experiments and datasets.

In addition, we addressed the reviewer's concerns about the signal to noise ratio of our ChIP. We show new heatmaps comparing the ChIP signals versus Input libraries for each replicate (see Figures 3, 4, 5 for reviewer below and **new Supplementary Fig. 7f**). In all cases, the signal is specific and only seen in the ChIP samples and therefore very unlikely to have been caused by noise.

2. The authors should more stringently assess the individual biological replicates (each of the 30 cycle pair versus each of the 36 cycle pair versus the merge - a 5 way comparison).

a. The authors should call peaks on their individual biological replicates from the 30 and 36 cycles (not merged) using the same parameters used in the current manuscript and compare these to each other and the peaks from the current ones derived from the merged datasets. Per the methods, it appears 2 replicates per condition (30, 36 cycles) were generated and merged. Are the peaks the same across all the replicates compared to the merged?

RESPONSE:

We would like to clarify our approach described in the manuscript. We called peaks in individual replicates of WFS1 30 and 36 cycles separately, and then merged these peaks and used them as our peak set for further analyses. In order to show the peaks called in the individual ChIP replicates (WFS1, FLAG Tag WFS1, 30 and 36 sonication cycles each) we extended the table shown in **new Supplementary Fig. 7e**, now including also peak calling data from FLAG Tag WFS1 ChIP, generated using the exact same parameters as used for WFS1 peaks.

Overall, the number of peaks and the total peak length were similar for the replicates (WFS1 and FLAG Tag WFS1). The number of peaks was generally lower in 36 versus 30 sonication cycles ChIP. This is expected, because, as outlined in the manuscript (page 16) the 36 cycles ChIP samples are enriched in heterochromatin (containing fewer active enhancers), while the 30 cycles ChIP samples represent mostly open chromatin. Thus, in order to include all WFS1 binding sites within both euchromatin- and heterochromatin-enriched samples we performed all subsequent analyses using merged peak sets (30 plus 36 cycles peaks) in the manuscript.

For the reviewer, we show a comprehensive analysis of the individual peak sets in the 30 and 36 sonication cycles ChIP in both replicates (WFS1 and FLAG Tag WFS1 ChIP) and compared these to the merged peak sets (30 + 36 cycles peaks combined).

In particular, we generated heatmaps showing the WFS1 and FLAG Tag WFS1 signals from both the 30 and 36 sonication cycles ChIP on merged peaks from both replicates (see below, Figure 1 for reviewer).

Additionally, we show heatmaps for the WFS1 and FLAG Tag WFS1 30 and 36 sonication cycles signals on peaks called individually from 30 and 36 sonication cycles ChIPs (see below, Figure 2 for reviewer).

Overall, these analyses reveal similar ChIP signals in the replicates for both sonication cycles, supporting the specificity and reproducibility of the WFS1 ChIP.

b. The authors should compare the signal intensity specifically at called peaks in the merge to signal at the same location across all 4 replicates. These data can be plotted as a meta plot - merged and each individual replicate.

RESPONSE:

We generated heat maps showing the enrichment of WFS1 and FLAG Tag WFS1 in 30 and 36 sonication cycles ChIPs on merged peaks (30 plus 36 cycles) across the replicates (Figure 1 for reviewer).

We included this analysis also as **new Supplementary Fig. 7g in the manuscript.**

Figure 1 for reviewer: Average binding profiles and heat maps showing enrichment of WFS1 and FLAG Tag WFS1 signal (log₂[ChIP/input]) in individual ChIP experiments using 30 and 36 cycles of sonication on merged WFS1 peaks (30 plus 36 cycles) ± 0.5 kb from center. Heat maps are ranked according to WFS1 enrichment in descending order.

In addition, we show here for the reviewer the ChIP signal of 30 and 36 sonication cycles samples of both replicates (WFS1 and FLAG Tag WFS1) on peaks called individually from these samples (Figure 2 for reviewer).

WFS1 log₂ (ChIP/Input) on CHIP-seq peaks

Figure 2 for reviewer: Average binding profiles and heat maps showing enrichment of WFS1 and FLAG Tag WFS1 ($\log_2[\text{ChIP}/\text{input}]$) in individual CHIP experiments using 30 and 36 cycles of sonication on individual WFS1 and FLAG Tag WFS1 peaks (30 and 36 cycles separately) ± 0.5 kb from center. Heat maps are ranked according to WFS1 enrichment in descending order.

3. The individual replicate input tracks are not provided. If the authors called peaks on the input replicates using the same parameters used in the current manuscript- how often do the peaks overlap with the ones identified in the WFS1 ChIP-seq?

RESPONSE:

We cannot call peaks from input signal only, as the peak calling algorithm uses the ChIP signal over the input signal to generate peaks.

In order to respond to the reviewer's suggestion, we generated heatmaps of the RPKM normalized (Reads Per Kilobase per Million mapped reads) signal in WFS1 and FLAG Tag WFS1 ChIP samples on merged peaks and compared these with the input signal on the same peak regions. These data clearly show that the ChIP signal is specifically enriched on called peaks, while the inputs show background signal only (see Figures 3 and 4 for reviewer).

We added the RPKM signal heatmap on WFS1 ChIP as **new Supplementary Fig. 7f**.

Figure 3 for reviewer: Average binding profiles and heat maps showing enrichment of WFS1 signal (RPKM) in individual ChIP experiments using 30 and 36 cycles of sonication on WFS1 merged peaks (30 plus 36 cycles) and on the same regions in respective input samples ± 0.5 kb from center. Heat maps are ranked according to WFS1 enrichment in descending order.

ChIP signal coverage on merged ChIP-seq peaks

Figure 4 for reviewer: Average binding profiles and heat maps showing enrichment of WFS1 and FLAG Tag WFS1 ChIP signal (RPKM) in individual ChIP experiments using 30 and 36 cycles of sonication on merged peaks (30 plus 36 cycles) and on same regions in respective input samples \pm 0.5 kb from center. Heat maps are ranked according to WFS1 enrichment in descending order.

In addition, we show here for the reviewer the WFS1 and FLAG Tag WFS1 RPKM signals in 30 and 36 sonication cycles ChIPs on peaks individually called from 30 and 36 cycles samples as well as the respective input samples (Figure 5 for reviewer).

Overall, these analyses clearly show the specificity of the signal on the called peaks in ChIP versus input samples and address the concerns on signal to noise ratio raised by the reviewer.

Figure 5 for reviewer: Average binding profiles and heat maps showing enrichment of WFS1 signal and FLAG Tag WFS1 signal (RPKM) in individual ChIP experiments using 30 and 36 cycles of sonication on peaks individually called in 30 and 36 sonication cycles ChIP, and on same regions in respective input samples ± 0.5 kb from center. Heat maps are ranked according to WFS1 enrichment in descending order.

4. Metaplots of the merged and individual replicates should be provided showing signal at the same regions shown on Figure 7A.

RESPONSE:

Figure 7a shows heatmaps of the WFS1 signal ($\log_2[\text{ChIP}/\text{input}]$) in the ChIP following 30 sonication cycles sample on epigenetic marks. We added this info in the Figure legend.

For the reviewer, we provide here the signals of the WFS1 and FLAG Tag WFS1 ChIP replicates (30 and 36 sonication cycles each) on the same regions (peaks of epigenetic marks acquired from CISTROME database). Note that FLAG Tag WFS1 36 sonication cycles sample has the lowest signal, most likely due to enrichment of inactive heterochromatic regions in the 36 versus 30 cycles preparation (see also response to reviewer's point 2). However, the overall signal pattern is the same across samples (Figure 6 for reviewer).

WFS1 log₂ (ChIP/Input) on epigenetic modifications

In addition, we show here for the reviewer RPKM signals of epigenetic marks (acquired from CISTROME database) on the WFS1 and FLAG Tag WFS1 merged peaks in 30 and 36 cycles (Figure 7 for reviewer), revealing similar signal patterns.

These analyses support the specificity of our ChIP seq analyses in our replicates.

Figure 7 for reviewer: Average binding profiles and heat maps showing enrichment of ATAC-seq, H3K27ac, H3K4me1, H3K4me3, H3K27me3, and two different H3K9me3 ChIP seq signals (RPKM) on merged (30 and 36 cycles) WFS1 and FLAG Tag WFS1 ChIP seq peaks, \pm 0.5 kb from center. Heat maps are ranked according to signal enrichment in descending order.

5. The authors should provide a browser session with the raw and processed individual replicates and merged data displayed. The data from GEO is not available to be downloaded.

RESPONSE:

Our raw ChIP-seq data are available on **NCBI GEO with accession code GSE253460** – available for download only using the **reviewer token: qtkvaisylpspdgp** until the public release, (<https://www.ncbi.nlm.nih.gov/geo/query/acc.cgi?acc=GSE253460>).

Additionally, we set up a UCSC Genome Browser session showing RPKM signals of the input, WFS1 and FLAG-WFS1 ChIP, 30 and 36 sonication cycles each, as well as ($\log_2[\text{ChIP}/\text{input}]$) signals and peaks (individual 30 and 36 cycles, and merged peaks).

Please see our processed data on the UCSC Genome Browser by clicking on the link:

<https://genome.ucsc.edu/cgi-bin/hgTracks?hubUrl=https://metazoa.csb.univie.ac.at/data/NEO4J/fatih/hubWFS1/hub.txt&genome=mm10>

In addition, we show 5 example genomic regions on different chromosomes where similar signals in all replicates lead to similar peaks. These data can be accessed via the links shown below:

Please note that the log₂ ratio and RPKM tracks are shown in the autoscale mode by default, and input and ChIP RPKM samples can be best compared by showing these tracks at the same maximum signal scale, which can be adjusted using the settings available in the UCSC browser by clicking right on each track.

<https://genome.ucsc.edu/cgi-bin/hgTracks?hubUrl=https://metazoa.csb.univie.ac.at/data/NEO4J/fatih/hubWFS1/hub.txt&genome=mm10&position=chr2%3A121665000-121700000>

<https://genome.ucsc.edu/cgi-bin/hgTracks?hubUrl=https://metazoa.csb.univie.ac.at/data/NEO4J/fatih/hubWFS1/hub.txt&genome=mm10&position=chr19%3A44391000-44398000>

<https://genome.ucsc.edu/cgi-bin/hgTracks?hubUrl=https://metazoa.csb.univie.ac.at/data/NEO4J/fatih/hubWFS1/hub.txt&genome=mm10&position=chr7%3A47027500-47036000>

<https://genome.ucsc.edu/cgi-bin/hgTracks?hubUrl=https://metazoa.csb.univie.ac.at/data/NEO4J/fatih/hubWFS1/hub.txt&genome=mm10&position=chr1%3A5345000-5351000>

<https://genome.ucsc.edu/cgi-bin/hgTracks?hubUrl=https://metazoa.csb.univie.ac.at/data/NEO4J/fatih/hubWFS1/hub.txt&genome=mm10&position=chr9%3A47572500-47578000>

Below, we show a screen shot of our UCSC Genome Browser session in one of these genomic regions (Figure 8 for reviewer).

Figure 8 for reviewer: Screenshot of our UCSC Genome Browser session on a genomic region (chr2:121,665,000-121,700,000) depicting FLAG Tag WFS1 and WFS1 ChIP signals (RPKM) and respective input signals (RPKM) for 30 and 36 sonication cycles samples, as well as ChIP signals ($\log_2[\text{ChIP}/\text{input}]$) and individual and merged peaks.

6. Fig 9E - the signal at “non-bound” enhancers should be compared to “bound regions” (and ideally permuted) - not random regions across the genome.

RESPONSE:

As requested, we changed **Supplementary Fig. 9e**, now showing heatmaps of WFS1 enrichment (30 and 36 sonication cycles) on overlapping (bound) versus non-overlapping (non-bound) enhancers rather than versus randomized regions.

In addition, for the reviewer we show here also the enrichment on WFS1 signal together with FLAG Tag WFS1 signal on the overlapping versus non overlapping enhancers in ChIP experiments using 30 and 36 sonication cycles (Figure 9 for reviewer). These analyses reveal similar signal patterns, again supporting specificity and interpretability of the ChIP.

WFS1 log2 (ChIP/Input)

Figure 9 for reviewer Average binding profiles and heat maps displaying enrichment of WFS1 and FLAG Tag WFS1 ($\log_2[\text{ChIP}/\text{input}]$) on predicted enhancer regions inside and outside of WFS1 peaks $\pm 3\text{kb}$ from center. Heat maps are ranked according to WFS1 enrichment in descending order.

Other residual concerns:

7. The authors' manuscript title is "Peripheral localization of the Myod1 gene....", but - as mentioned in my original review - the probe in Fig 3 is infrequently at the periphery. Thus, it is not relevant that occasionally LADs are not at the periphery. Moreover, given the title, it makes it hard for me to understand how lack of peripheral positioning makes it an applicable locus to support their title.

RESPONSE:

We agree with the reviewer that not all LADs are at the periphery in all cells in a mixed population. Kind et al.¹ showed that in a given cell only 30% of all LADs are at the periphery and that with every cell cycle LADs are reshuffled within nuclear space. We compared the distributions of distances of LADs from the nuclear periphery in a mixed cell population and found statistically significant difference in *Lmna/LBR* double knockout versus WT cells. This is a common way to compare positions of genes/loci within the nuclear space (see manuscript by Shachar et al.² and Figure 10 for reviewer below).

However, we don't know what is meant by the reviewer's comment on the title "Peripheral localization of *MyoD1*....." and how this relates to the position of LADs shown in Fig. 3. The FISH data in Fig. 3 are to show that peripheral localization of LADs is influenced by the lamin A/C and LBR KO, while *MyoD1* is not affected. Both, localization of LADs and of *MyoD1* are analyzed by distribution curves (Figure 11 for reviewer),

[Figure redacted]

Figure 10 for reviewer: Figure shows data from Shachar et al.², comparing the localization of a LAD, a highly transcribed locus (expected to be in the interior), and an "in-between" locus (from left to right).

Figure 11 for reviewer. Figure shows data in Figure 3g represented as distributions curves instead of violin plots to illustrate the shift of the localization of the LAD region towards the nuclear interior upon depletion of LA/C and LBR.

8. Please list the H3K9me2 ab used in the methods.

RESPONSE:

The H3K9me2 antibody used in immunofluorescence is listed in the supplementary materials & methods table.

Histone ChIP data were removed from the manuscript, as we consider them preliminary and results are not relevant for the conclusions of the paper (see below). Thus, H3K9me2 antibody is no longer mentioned in methods section (page 36).

9. Are the cells cross linked in the experiments in Supp Fig 7D? It is unclear from the figure legend. Also, an input is not included.

RESPONSE:

We now add in the legend the information that we used the same conditions for Western blots as for ChIP, thus the samples are cross-linked. As for the input, the signal is usually very weak and hardly detectable (see also Western blot of non-cross-linked samples in Figure 5c). However, the use of WFS1 KO samples provides an ideal control for the antibody.

10. SFig 3G (Left) and SFig 8D - the experiment should be repeated with biological replicates and not technical.

RESPONSE:

As first and co-corresponding authors of the manuscript have left the lab and could not repeat all the requested experiments, we had to prioritize. We generated additional biological replicates for ChIP-qPCR analyses in Tmem38a knockout cells. New

Supplementary Fig. 8c shows loss of WFS1 enrichment on *MyoD1* enhancer in *Tmem38a* knockout compared to wildtype cells. This result is in line with the observed release of the *MyoD1* locus from the nuclear periphery in *Tmem38a* knockout versus wildtype cells.

In contrast, we consider the results of histone ChIPs on the *MyoD1* promoter (the old Supplementary Fig. 3g) as preliminary. In order to make a conclusive analysis of epigenetic marks on the *MyoD1* locus, it would require many more additional experiments beyond the scope of this study. As we mentioned these preliminary data only as a side aspect in the original manuscript, we decided to remove this Figure and relevant information from the manuscript completely.

11. Fig 5C - is the WT:NET39 KO comparison statistically different?

RESPONSE:

As indicated in Figure 4b the WT:NET39 KO comparison of the normalized data is not significant. We have now added the bar with ns, also in Supplementary Fig. 5c, where the comparison of absolute distance data results in $p_{\text{NET39 KO}}=0.5$, as indicated in the Figure legend.

Reviewer #3 (Remarks on code availability):

I do not have expertise to review code.

Reviewer #4 (Remarks to the Author):

I did not review the initial version of this manuscript, but have been asked to evaluate the authors' response to reviewer 1. After reading of the manuscript and the response, I find that the authors have diligently addressed the original points by reviewer 1. The same applies to their responses to the other reviewers. While there are some loose mechanistic ends, the reported findings point to a novel mechanism of gene positioning and will be of interest to many in the field.

RESPONSE:

We thank the reviewer for her/his positive evaluation of our revisions.

Reviewer #4 (Remarks on code availability):

I am not qualified to review code.

References

- 1 Kind, J. *et al.* Single-Cell Dynamics of Genome-Nuclear Lamina Interactions. *Cell* **153**, 178-192 (2013). <https://doi.org/10.1016/j.cell.2013.02.028>
- 2 Shachar, S., Voss, T. C., Pegoraro, G., Sciascia, N. & Misteli, T. Identification of Gene Positioning Factors Using High-Throughput Imaging Mapping. *Cell* **162**, 911-923 (2015). <https://doi.org/10.1016/j.cell.2015.07.035>

Other things:

We added a statement on whether sex was included in our study in the Reporting Summary file.

FINAL SUBMISSION, Georgiou et al.

We are pleased to see that reviewer 3 appreciates our efforts to address the specificity of the ChIP data. Following her/his suggestion we now add a sentence in the discussion acknowledging a certain degree of noise in our ChIP signal as a technical limitation of this study.

In addition, we addressed all editorial requests.

POINT-BY-POINT RESPONSE TO REVIEWER'S COMMENT

Reviewer #3 (Remarks to the Author):

I appreciate the authors efforts and additional experiments to address my concerns and the new data that is provided. Examining the data on the genome browser link provided by the authors suggests some degree of noise - at least in part likely driven by the double cross linking required to examine WFS1 enrichment. This is also consistent with the residual ChIP-qPCR signal observed in the KO cells. I suggest that the authors include this point as a technical limitation in the discussion.

RESPONSE:

We thank the reviewer for appreciating our efforts to address her/his remaining concerns and for all her/his suggestions during the entire review process to show specificity and reproducibility of the ChIP data. We agree that there is some degree of noise in our WFS1 ChIP, but we would like to point out that with all the controls for ChIP, now included in the final manuscript (in large parts requested by this reviewer) the data show specific enrichment of WFS1 on a subset of active enhancers supporting the main conclusions of the paper.

We follow the suggestion of the reviewer and added a note on the signal noise of our ChIP data as a technical limitation of our study in the discussion section on page 21, end of first paragraph.